# FUS affects circular RNA expression in murine embryonic stem cell-derived motor neurons

Lorenzo Errichelli[1,2,*], Stefano Dini Modigliani[1,*], Pietro Laneve[1], Alessio Colantoni[2], Ivano Legnini[2], Davide Capauto[1,2], Alessandro Rosa[1,2], Riccardo De Santis[1,2], Rebecca Scarfò[2], Giovanna Peruzzi[1], Lei Lu[3], Elisa Caffarelli[1,4], Neil A. Shneider[3], Mariangela Morlando[2] & Irene Bozzoni[1,2,4,5]

The RNA-binding protein FUS participates in several RNA biosynthetic processes and has been linked to the pathogenesis of amyotrophic lateral sclerosis (ALS) and frontotemporal dementia. Here we report that FUS controls back-splicing reactions leading to circular RNA (circRNA) production. We identified circRNAs expressed in in vitro-derived mouse motor neurons (MNs) and determined that the production of a considerable number of these circRNAs is regulated by FUS. Using RNAi and overexpression of wild-type and ALS-associated FUS mutants, we directly correlate the modulation of circRNA biogenesis with alteration of FUS nuclear levels and with putative toxic gain of function activities. We also demonstrate that FUS regulates circRNA biogenesis by binding the introns flanking the back-splicing junctions and that this control can be reproduced with artificial constructs. Most circRNAs are conserved in humans and specific ones are deregulated in human-induced pluripotent stem cell-derived MNs carrying the FUS[P525L] mutation associated with ALS.

[1] Center for Life Nano Science@Sapienza, Istituto Italiano di Tecnologia, Viale Regina Elena 291, Rome 00161, Italy. [2] Deparment of Biology and Biotechnology 'Charles Darwin', Sapienza University of Rome, P.le A. Moro 5, Rome 00185, Italy. [3] Department of Neurology, Center for Motor Neuron Biology and Disease, Columbia University, 630 W 168th Street, New York, New York 10032, USA. [4] Institute of Molecular Biology and Pathology, CNR, Sapienza University of Rome, P.le A. Moro 5, Rome 00185, Italy. [5] Institute Pasteur Fondazione Cenci-Bolognetti, Sapienza University of Rome, P.le A. Moro 5, Rome 00185, Italy. * These authors contributed equally to this work. Correspondence and requests for materials should be addressed to M.M. (email: mariangela.morlando@uniroma1.it) or to I.B. (email: irene.bozzoni@uniroma1.it).

A new class of covalently closed circular RNA molecules (circRNAs) has recently become the object of intensive study. First described as rare events[1–3], recent studies have demonstrated that circRNAs are commonly produced by thousands of genes from Archaea to mammals[4–6]. Interestingly, in higher eukaryotes they are highly expressed in neuronal tissues and enriched at synapses, suggesting a specific involvement in neuronal processes[7,8]. Moreover, circRNAs are more abundant than their host gene linear mRNA isoforms in the neuropil and dendrites, suggesting that they may regulate synaptic function and neuronal plasticity[8].

So far, very little is known about their function: some can act as sponges for microRNAs and proteins[9–11] or can compete with linear RNA production regulating the accumulation of full-length mRNA[12]. CircRNAs also regulate transcription of their parental genes by association with the RNA polymerase II machinery[13]. Notably, emerging data point to a potential role of circRNAs in human diseases[14,15], with clear evidence of tumor-promoting properties in in vivo models[16]. In the nervous system, the best-studied circRNA, CDR1, was found expressed in neocortical and hippocampal neurons and downregulated in Alzheimer disease[17]. Through its ability to sponge miR-7 (refs 9,10), it could play a crucial role in nervous system diseases deregulating targets with important function[17,18].

CircRNAs originate from a back-splicing reaction in which a downstream 5′ splice site interacts with an upstream 3′ splice site, leading to the formation of a covalently closed circRNA[19]. The mechanisms underlying these events are not fully understood; however, in mammals it has been shown that complementarity between inverted sequences inside flanking introns[3,20–22] and the activity of RNA-binding proteins (RBPs)[12,23] enhance the juxtaposition of the splice sites involved in the back-splicing reaction. Muscleblind, a splicing factor derived from the Mbl gene, was the first example of an RBP controlling the levels of the circRNA derived from its second exon by binding both flanking introns[12]. Afterwards, Quaking (QKI), a splicing factor that promotes myelination and oligodendrocyte differentiation[24,25], was also described as a circRNA regulator[23]. Finally, many hnRNPs as well as SR proteins are involved in circRNA production in flies[26].

The RBP FUS has a well-characterized role in splicing regulation[27] with several splicing factors identified as FUS interactors[28–31]. FUS functions are particularly interesting since several mutations have been causally linked to amyotrophic lateral sclerosis (ALS)[32,33]. Most ALS-linked FUS mutations cluster in the C-terminus of the protein in or near the nuclear localization signal. This leads to the mislocalization of the protein to the cytoplasm, with decrease of FUS levels in the nucleus and formation of abnormal cytoplasmic aggregates[32–34]. Aberrant RNA metabolism due to FUS mutations by gain- and/or loss-of-function has been proposed as a key mechanisms in the pathogenesis of ALS and frontotemporal dementia[35]; moreover, deregulation of splicing has been linked to several neurological diseases[32,36,37].

In this study, we identify circRNAs expressed in in vitro-derived motor neurons (MNs) and we analyse whether FUS may be involved in the control of back-splicing events leading to circRNA formation. We characterize several circRNAs that are affected by FUS depletion and by FUS mutations associated with familial forms of ALS. Notably, for selected circRNAs, we demonstrate the enrichment of FUS binding on circularizing exon–intron regions by cross-linking immunoprecipitation (CLIP) and the direct role of the protein in regulating back-splicing. Finally, most of these circRNAs are expressed in induced pluripotent stem cells (iPSCs)-derived human MNs and two of them undergo similar FUS-dependent regulation

in ALS-associated $FUS^{P525L}$ genetic background. Altogether, our data suggest a possible conserved function of this novel class of transcripts and provide an interesting link with the ALS pathology.

## Results

**Identification of circRNAs in mESC-derived MNs.** Mouse embryonic stem cells (mESCs), derived from wild-type ($FUS^{+/+}$) or knock out ($FUS^{-/-}$)[38] FUS mice and expressing a green fluorescent protein (GFP) reporter under the control of the MN-specific Hb9 promoter (Hb9::GFP transgene)[39], were differentiated into bona fide MNs according to Wichterle et al.[40] (Supplementary Fig. 1a). In agreement with this procedure, Pax6 and Olig2 transcription factors, responsible for establishing MN progenitors, were found in the Hb9::GFP⁻ cells while genes required for consolidation of MN identity (Hb9) and for development (Islet-1) and function (ChAT) of spinal MNs were highly enriched in Hb9::GFP⁺ cells. As expected, the markers for astrocytes (Gfap) and oligodendrocytes (Pdgfr-α) were almost undetectable in both cell populations as well as the V1, V2 (Bhlhe22) and V3 (Sim1) interneuron markers (Supplementary Fig. 1b,c). Total RNA from purified GFP⁺-$FUS^{+/+}$ and GFP⁺-$FUS^{-/-}$ MNs was sequenced by ribo-Zero Next-Generation Sequencing from three biological replicates. A dedicated pipeline for in silico circRNA detection was then applied (find_circ)[10] to identify circRNAs and to evaluate their expression levels. Briefly, reads mapping to ribosomal and other abundant non-coding RNAs were discarded (see Methods section), as were reads mapping contiguously to the reference genome, and the unmapped reads were used as input for circRNA identification (Fig. 1a). Since no reference transcriptome is used in the procedure, the back-splicing sites of the identified circRNAs do not necessarily coincide with annotated splice sites. The number of reads mapping on back-splicing and on corresponding linear-splicing junctions was computed. Three thousand nine hundred and eighty circRNAs were identified, having at least two unique reads mapping on their back-splicing junction in at least one sample (Table 1). This number is similar to that obtained from sequencing experiments previously performed on other neuronal samples[7], confirming the high abundance of circRNAs in neuronal tissues, now also including in vitro mESC-derived MNs. We identified 3,894 circRNAs within the body of 2,097 known genes, many hosting more than one circRNA. As shown in Fig. 1b, the vast majority of these genes are protein-coding. Analysing the localization of circRNAs within the body of protein-coding transcripts (Supplementary Fig. 1d), we found that most of them are fully included in the coding region with a proportion spanning across the 5′ untranslated region higher than expected (22%, P value for chi-squared test = $1.15e^{-28}$) (Fig. 1c and Supplementary Fig. 1e).

**CircRNA expression is modulated in $FUS^{-/-}$ MNs.** To retain only those circRNAs that are robustly expressed (two or more unique reads in at least three samples), the identified candidates were then filtered to generate a list of 1,153 circRNAs from a total of 785 hosting genes (Supplementary Data 1). Using this list, we performed a differential expression analysis in $FUS^{-/-}$ versus $FUS^{+/+}$ conditions (see Methods section), and we plotted the log2 fold change values of the 1,153 circRNAs against those of their cognate linear transcripts, whose expression was quantified using the reads assigned to the linear junctions (Fig. 1d). In $FUS^{-/-}$ condition, we found a general reduction in the expression of circRNAs (P value for one-sample t-test = $3.16e^{-24}$), while the distribution of fold changes of their cognate linear RNAs is centered at 0 (P value for one-sample t-test = 0.27) (Fig. 1d).

The differential expression analysis revealed that, in the absence of FUS, the expression level of 136 circRNAs varied significantly (Supplementary Data 2). Notably, 111 out of the 136 circRNAs were downregulated in $FUS^{-/-}$ MNs (Fig. 1d and Supplementary Data 2). For the majority of the dysregulated circRNAs, changes were not concordant with their linear

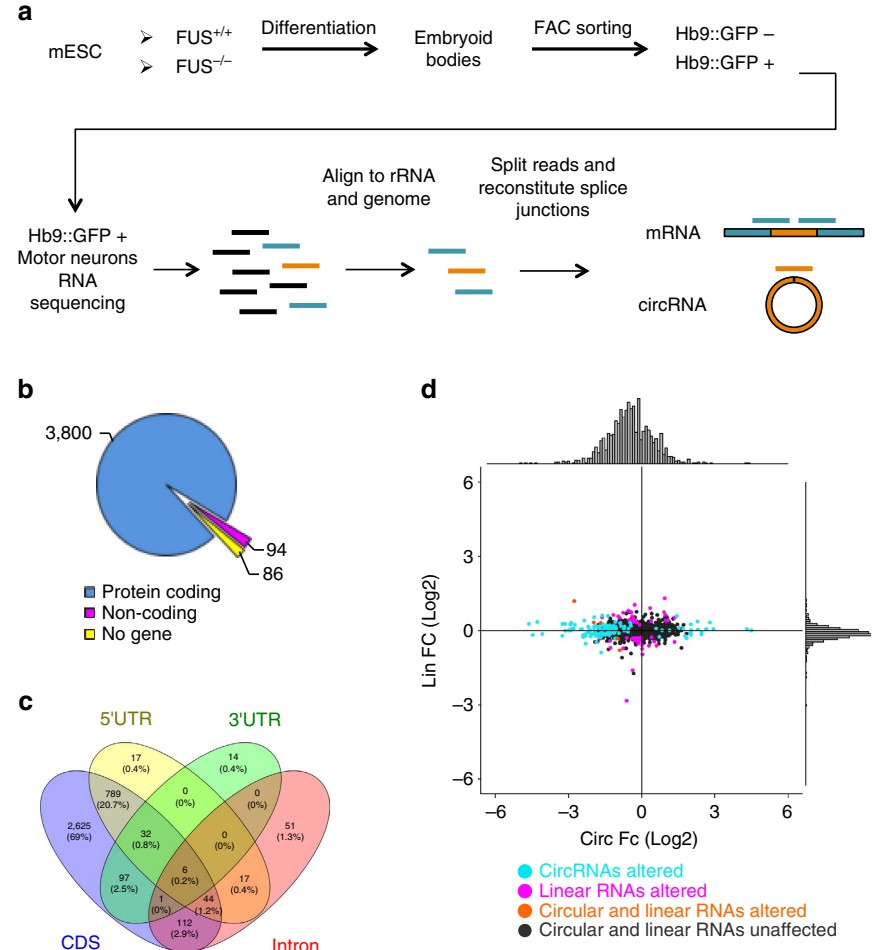

**Figure 1 | mESC _in vitro_ differentiation to spinal motor neurons and RNA-seq analysis.** (**a**) Schematic overview of the murine motor neuron differentiation protocol and of the RNA-seq analysis. The procedures are reported in Supplementary Fig. 1a and in Methods section. (**b**) Pie chart showing the number of circRNAs hosted by protein-coding, non-coding genes and intergenic regions (No gene). (**c**) Venn diagram showing circRNA localization within the body of protein-coding genes. Numbers refer to circRNA species for each category. (**d**) Scatter plot showing the correlation of log2 fold change of circRNAs (x axis) and their cognate linear RNAs (y axis) in $FUS^{-/-}$ conditions. The distributions of the fold change values are shown above and aside the scatter plot. Turquoise dots indicate when only circRNA species are altered in $FUS^{-/-}$; magenta dots indicate when only linear RNAs are deregulated; circRNA and linear counterparts either deregulated or unaffected are indicated by orange and black dots, respectively.

**Table 1 | Summary of RNA-seq results for six different samples.**

| Genotype | $FUS^{+/+}$ | | | $FUS^{-/-}$ | | |
|---|---|---|---|---|---|---|
| Samples | Replicate 1 | Replicate 2 | Replicate 3 | Replicate 1 | Replicate 2 | Replicate 3 |
| Raw read pairs | 44,248,846 | 41,761,535 | 54,227,816 | 42,627,671 | 42,543,241 | 57,617,180 |
| Read pairs after preprocessing | 42,373,468 | 39,810,100 | 52,093,794 | 40,626,338 | 39,716,300 | 55,182,405 |
| Read pairs after abundant transcript filtering | 38,203,006 | 34,387,987 | 46,426,839 | 36,306,476 | 35,475,603 | 49,580,705 |
| Reads not mapping linearly to genome | 14,212,137 | 13,267,421 | 17,151,574 | 14,354,276 | 13,429,149 | 17,228,985 |
| Back-splicing reads unfiltered | 30,153 | 30,408 | 35,905 | 27,006 | 25,867 | 32,829 |
| Linear splicing reads unfiltered | 8,325,574 | 7,776,167 | 10,075,345 | 8,458,768 | 7,954,743 | 10,095,981 |
| Back-splicing reads | 15,303 | 18,064 | 19,548 | 14,704 | 13,854 | 16,584 |
| Linear splicing reads | 8,249,195 | 7,704,546 | 9,984,445 | 8,384,786 | 7,882,403 | 10,005,241 |
| Back-splicing junctions | 6,687 | 7,345 | 7,793 | 6,413 | 6,100 | 6,943 |
| Linear splicing junctions | 183,464 | 175,357 | 189,769 | 186,635 | 182,302 | 193,240 |
| Circular RNAs, minimum two reads | 1,299 | 1,675 | 1,624 | 1,280 | 1,235 | 1,399 |
| Total circular RNAs | 4,076 | | | | | |
| Total circular RNAs not spanning multiple loci | 3,980 | | | | | |

**Table 2 | List of circRNAs analysed and their linear counterparts.**

| Name | ID | Coordinates (GRCm38) | Host gene name | CircRNAs | | | | Linear RNAs | | | |
|---|---|---|---|---|---|---|---|---|---|---|---|
| | | | | c.p.m. in $FUS^{+/+}$ | c.p.m. in $FUS^{-/-}$ | Fold change | $P$ value | c.p.m. in $FUS^{+/+}$ | c.p.m. in $FUS^{-/-}$ | Fold change | $P$ value |
| c-01 | circ_3279 | 1:89604598 − 89634294 | Agap1 | 6.871 | 1.485 | − 2.163 | 0.008 | 101.960 | 95.095 | − 0.096 | 0.564 |
| c-02 | circ_1223 | 13:59454535 − 59546383 | Agtpbp1 | 7.343 | 1.902 | − 1.884 | 0.010 | | | | |
| c-03 | circ_5846 | 8:122908668 − 122916045 | Ankrd11 | 10.720 | 4.836 | − 1.086 | 0.066 | 115.098 | 135.393 | 0.232 | 0.129 |
| c-13 | circ_0273 | 10:93221587 − 93239043 | Cdk17 | 8.540 | 1.258 | − 2.500 | 0.001 | 148.740 | 153.840 | 0.043 | 0.807 |
| c-16 | circ_5906 | 8:36937560 − 36938758 | Dlc1 | 58.000 | 87.875 | 0.592 | 0.009 | 377.461 | 397.579 | 0.075 | 0.591 |
| c-27 | circ_1165 | 13:36232122 − 36246494 | Fars2 | 6.709 | 1.258 | − 2.178 | 0.014 | 40.650 | 43.657 | 0.110 | 0.647 |
| c-31 | circ_5722 | 7:81893799 − 81905636 | Hdgfrp3 | 89.831 | 53.936 | − 0.741 | 0.002 | 315.508 | 323.305 | 0.035 | 0.743 |
| c-45 | circ_5313 | 6:5100432 − 5115480 | Ppp1r9a | 5.818 | 1.774 | − 1.592 | 0.051 | 57.456 | 58.201 | 0.012 | 0.952 |
| c-48 | circ_1733 | 15:39566943 − 39616510 | Rims2 | 69.158 | 32.256 | − 1.091 | 0.000 | 83.871 | 67.287 | − 0.322 | 0.075 |
| c-52 | circ_5622 | 7:46590607 − 46635556 | Sergef | 8.549 | 1.387 | − 2.458 | 0.004 | 43.857 | 45.445 | 0.071 | 0.763 |
| c-75 | circ_0907 | 12:51621780 − 51661713 | Strn3 | 17.591 | 4.744 | − 1.896 | 0.000 | 123.639 | 119.365 | − 0.044 | 0.756 |
| c-76 | circ_0919 | 12:52516079 − 52519967 | Arhgap5 | 91.143 | 55.960 | − 0.700 | 0.006 | 61.603 | 68.884 | 0.163 | 0.318 |
| c-77 | circ_0920 | 12:52516079 − 52542636 | Arhgap5 | 19.451 | 10.481 | − 0.870 | 0.026 | 37.840 | 46.675 | 0.312 | 0.126 |
| c-78 | circ_2111 | 16:94383912 − 94393174 | Ttc3 | 83.164 | 26.259 | − 1.684 | 0.000 | 163.366 | 201.963 | 0.309 | 0.023 |
| c-79 | circ_2193 | 17:26736789 − 26743131 | Crebrf | 9.619 | 3.646 | − 1.385 | 0.019 | 14.501 | 21.935 | 0.582 | 0.074 |
| c-80 | circ_4484 | 4:48604581 − 48617331 | Tmeff1 | 96.109 | 43.729 | − 1.128 | 0.000 | 1,284.342 | 1,193.313 | − 0.107 | 0.316 |
| c-82 | circ_3217 | 1:64078707 − 64079334 | Klf7 | 1.745 | 8.033 | 2.029 | 0.005 | 134.317 | 135.387 | 0.012 | 0.944 |
| c-83 | circ_3834 | 2:74568941 − 74573626 | Lnp | 16.414 | 30.085 | 0.836 | 0.026 | 157.845 | 158.449 | 0.007 | 0.962 |
| c-84 | circ_2254 | 17:43037595 − 43040189 | Tnfrsf21 | 1.011 | 4.259 | 2.006 | 0.037 | 89.263 | 105.678 | 0.242 | 0.160 |
| c-87 | circ_5306 | 6:47576563 − 47577667 | Ezh2 | 21.829 | 35.535 | 0.721 | 0.043 | 26.833 | 22.479 | − 0.260 | 0.282 |
| c-88 | circ_1224 | 13:59460469 − 59462173 | Agtpbp1 | 0.467 | 3.547 | 2.628 | 0.012 | 161.150 | 166.469 | 0.045 | 0.772 |

counterpart, with only two of them significantly paralleling the behaviour of the host linear transcript (Fig. 1d and Supplementary Data 2). As circRNA dysregulation in these two cases could be due to transcriptional regulation of the host gene and not to a post-transcriptional event, these two circRNAs were excluded from further analysis.

Focussing on the 134 circRNAs that appeared to be regulated at the post-transcriptional level, we applied several criteria to restrict the number of species for a more refined molecular analysis. We selected circRNAs having at least two reads on average among all six samples sequenced, dysregulation of log2 fold change $> 0.4$ and localization in host genes with a known or potential role in neuronal physiology. This filtering allowed the selection of 21 circRNAs (Table 2) whose circularity was verified by RNaseR resistance (Supplementary Fig. 2a,b). C-76 and c-77 showed high sensitivity to RNaseR and therefore they were not considered for further analysis. For almost every circRNA, reverse transcription–PCR (RT–PCR) produced a band of the expected size in addition to longer products, possibly corresponding to concatemers, generated by rolling circle retro-transcription[5]. For two circRNAs, c-01 and c-87, short and long products were gel purified and sequenced, confirming that indeed they were back-splicing products and concatemers, respectively (Supplementary Fig. 2c,d). The expression of the 19 validated circRNA species was then measured by quantitative RT–PCR in $GFP^+$-$FUS^{+/+}$ and $GFP^+$-$FUS^{-/-}$ MNs and all of them resulted significantly deregulated, in agreement with RNA-seq: in $FUS^{-/-}$ conditions, 14 were downregulated and 5 upregulated (Fig. 2a). The modulation of the corresponding linear transcripts was also measured in $FUS^{-/-}$ versus $FUS^{+/+}$ conditions, and in all cases no significant variations were observed, indicating that the altered levels of the circular molecules were not due to transcriptional changes (Supplementary Fig. 2e).

In order to address the MN specificity of these species, we compared their levels in undifferentiated $FUS^{+/+}$ mESCs and in differentiated $GFP^-$-$FUS^{+/+}$ and $GFP^+$-$FUS^{+/+}$ cells

(Fig. 2b). The data indicate that several circRNAs are expressed already in proliferating mESCs, while others are upregulated during differentiation with nine species enriched in $GFP^+$ MNs (c-1, c-2, c-13, c-16, c-48, c-80, c-82, c-84 and c-88). Northern blot was also used as a means to analyse circRNA expression; for two abundant species, c-31 and c-78, the upregulation upon differentiation and the differential expression levels in $FUS^{+/+}$ versus $FUS^{-/-}$ MNs were confirmed (Fig. 2c).

Analysis of the expression of the linear counterparts during differentiation indicated that most of them showed a similar variation to the circular molecules. However, in the case of c-16 and c-79 an inverse correlation was found, with the linear forms (l–16 and l–79) being highly expressed in mESCs and down-regulated upon differentiation (Supplementary Fig. 2f).

To determine whether the expression of these selected circRNAs is conserved in human MNs, we generated MNs from a $FUS^{+/+}$ iPSC line[34] containing the $HB9::GFP$ reporter[41]. $GFP^+$ cells were isolated from a mixed population and analysed for the expression of MN-specific markers. These cells, while having undetectable amounts of the $NANOG$ pluripotency marker, exhibited high expression levels of $HB9$, $ISLET1$ and $CHAT$, consistent with their MN identity (Supplementary Fig. 2g). For each circRNA, divergent oligonucleotide pairs were designed on the human exons corresponding to those involved in the back-splicing events identified in mouse. Seventeen out of the 19 species were also detected in humans with some of them showing specific enrichment in the $GFP^+$ MN-enriched fraction (Supplementary Fig. 2h).

**FUS depletion and mutation affect the biogenesis of circRNAs.** To further analyse the FUS dependence of circRNA biogenesis, we investigated in the Neuro-2a (N2a) cell line the modulation of the 19 validated circRNAs in response to FUS downregulation or overexpression. N2a cells are neural crest-derived neuroblasts that respond quickly to serum deprivation and other stimuli (for example, retinoic acid (RA)) by expressing genes that lead to neuronal differentiation and neurite outgrowth[42,43]. All but one

circRNAs were found to be expressed in this cell line (Supplementary Fig. 3a). Nuclear/cytoplasmic fractionation of N2a cells demonstrated that 15 out of the 18 circular species were mainly localized in the cytoplasm, whereas the others (c-01, c-87 and c-88) had an almost exclusive nuclear localization (Supplementary Fig. 3b). Notably, sequencing of the nuclear

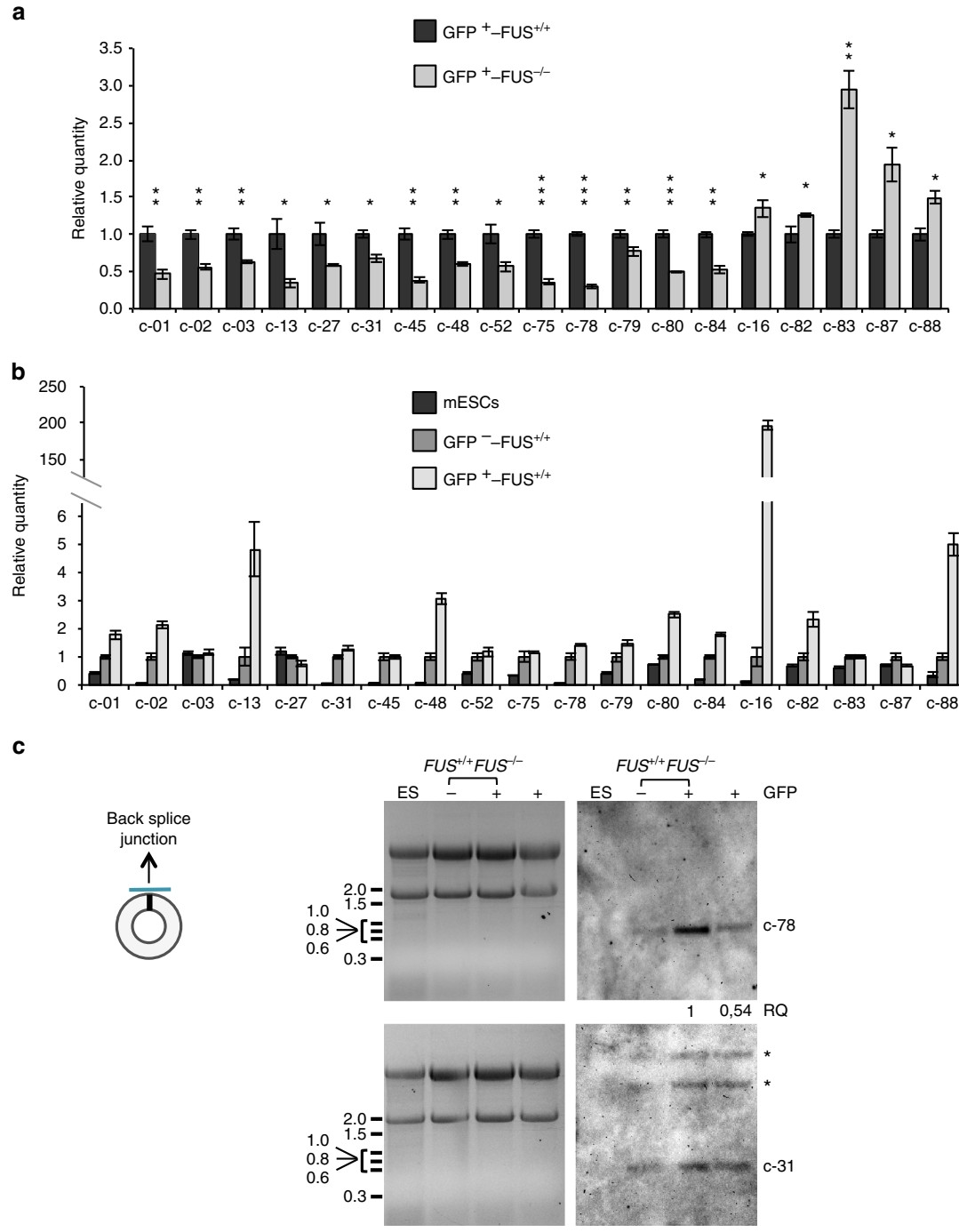

**Figure 2 | CircRNA expression upon FUS depletion and in MN differentiation of mESCs. (a)** circRNA expression analysed by qRT–PCR in sorted $GFP^+$-$FUS^{+/+}$ and $GFP^+$-$FUS^{-/-}$ cells. The 14 downregulated (left) and 5 upregulated (right) species are reported. CircRNA levels were normalized against Atp5o mRNA levels and expressed as relative quantity with respect to the $GFP^+$-$FUS^{+/+}$ sample set to a value of 1. **(b)** The histogram shows the expression level of circRNAs, measured by qRT–PCR, in $FUS^{+/+}$ mESCs and $GFP^+$-$FUS^{+/+}$ and $GFP^-$-$FUS^{+/+}$ cells. Their values were normalized against Atp5o mRNA levels and expressed as relative quantity with respect to $GFP^-$ samples set to a value of 1. For **a,b**, error bars represent s.e.m. of three independent experiments. *$P < 0.05$, **$P < 0.01$ and ***$P < 0.001$ correspond to two-tailed Student's $t$-test. **(c)** Left panel: location on the circRNA of the DIG-labelled probe used for northern blot analysis. Right panel: Northern blots on 10 µg of total RNA from $FUS^{+/+}$ mESCs, $GFP^-$-$FUS^{+/+}$, $GFP^+$-$FUS^{+/+}$ and $GFP^+$-$FUS^{-/-}$ cells. The circular forms are indicated aside the gels; asterisks indicate two additional bands likely corresponding to the two alternative linear forms of the c-31 host mRNAs (Hdgfrp3-001, ENSMUST00000107305.7, 5,887 nt; Hdgfrp3-002, ENSMUST00000026094.5, 2,865 nt). Each blot is shown in parallel to the EtBr staining of the agarose gel, where the migration of the 18S and 28S rRNAs is indicated.

species indicated the absence of intron sequences (Supplementary Fig. 2c,d); therefore, these circRNAs represent the first examples of completely spliced nuclear circRNAs[13].

We then tested the circRNA expression in differentiated N2a cells that were treated with siRNAs targeting the murine FUS (Fig. 3a,b). For 14 circRNAs, we observed a modulation concordant with that observed in $GFP^+$-$FUS^{-/-}$ MNs (Fig. 2a). In particular, 11 species were downregulated by FUS depletion, while 3 were upregulated (Fig. 3b). Notably, in no case concordant variation of the linear transcripts was observed, confirming that alteration of circRNAs was not due to effects on transcription (Supplementary Fig. 3c).

In the same system, we then analysed the effects of FUS rescue on circRNA production. N2a clones carrying Doxycycline (Dox)-inducible expression cassette for cDNAs encoding the human WT or the ALS-linked $FUS^{R521C}$ and $FUS^{P525L}$ mutant proteins were treated with siRNAs, unable to target the transgene, and subsequently with Dox. Comparable levels of exogenous FUS proteins were induced in the mutant clones with respect to the $FUS^{WT}$ (Fig. 3a and Supplementary Fig. 3d). Relative to $FUS^{WT}$, the $FUS^{R521C}$ and $FUS^{P525L}$ proteins were previously shown to decrease in the nuclear compartment and to be mislocalized to the cytoplasm, more so in the case of $FUS^{P525L}$ (refs 44,45). With respect to a control cell line carrying an empty vector (Ctrl), the ectopic expression of $FUS^{WT}$ was able to rescue the correct expression levels of almost all circRNAs (Fig. 3c). For those species that were downregulated in FUS RNAi, $FUS^{R521C}$ and $FUS^{P525L}$ failed to fully rescue circRNA levels with the strongest effect observed with $FUS^{P525L}$, the more mislocalized of the two mutant FUS proteins. Even if a simple hypothesis would correlate this phenotype with the amount of nuclear FUS, it cannot be excluded that the mutations per se lead to a loss of activity in splicing regulation; in fact, it was previously shown that both the $FUS^{R521C}$ and $FUS^{P525L}$ lead to decreased interactions with splicing promoting factors, such as the U1-70K (ref. 31). Therefore, by losing such interaction the mutant proteins could affect the proper utilization of specific splice junctions more sensitive to U1 snRNP recognition.

For the circRNAs upregulated upon FUS depletion, the $FUS^{R521C}$ and $FUS^{P525L}$ proteins were able to reduce circRNAs at the same levels as $FUS^{WT}$. These results can indicate that either low levels of nuclear FUS are sufficient to inhibit circularization or, alternatively, since both mutants were shown to have a higher binding affinity than the WT protein to the splicing-related factor SMN[31,46], they could act as stronger splicing inhibitors.

Therefore, through loss or gain of function, mutant FUS proteins can exert different role in splicing and back-splicing regulation depending on the protein–protein and RNA–protein complexes involved.

In order to test whether the FUS-dependent circRNA biogenesis observed in mouse is conserved in human, we analysed RNA from human iPSC-derived MNs carrying the P525L mutation either in heterozygous or homozygous form ($FUS^{WT/P525L}$ and $FUS^{P525L/P525L}$)[34]. Notably, two circRNAs downregulated upon FUS depletion were also downregulated in $FUS^{P525L/P525L}$ MNs, with no effect in the heterozygous (Fig. 3d). This very likely correlates with the fact that, in such experimental systems, it is impossible to reproduce the exact timing of FUS delocalization that occurs in ALS patients after several decades of life.

**FUS binds to circularizing exon–intron junctions**. Since the regulation of back-splicing normally involves the introns flanking the circularized exons, we preliminary retrieved and re-analysed public FUS CLIP-Seq data from the mouse brain[47] in search for FUS-binding sites within the 500 nt proximal to

back-splicing sites (1–500 nt regions) of the 134 circRNAs deregulated in $FUS^{-/-}$ MNs. We considered only those interactions where at least one CLIP-Seq peak mapped into one of the two flanking intronic regions. We noticed that the percentage of deregulated circRNAs that are bound by FUS (35%) is significantly higher than that of circRNAs unaffected by the lack of FUS (24%, P value for chi-squared test = 0.0208; Supplementary Fig. 4a). This enrichment was maintained when sequences up to 1,000 nucleotides were considered, while it was lost when higher distances were analysed (Supplementary Fig. 4a). These results are in agreement with the notion that the sequences involved in the control of back-splicing are in general in the proximal intron regions[12,21,22]. Notably, focussing on the 1–500 region, the percentage of deregulated circRNAs that are bound by FUS increases when a more stringent P value cutoffs is used to select differentially expressed circRNAs (0.001 or 0.005; Supplementary Fig. 4b); in other words, FUS preferentially binds the introns of those circRNAs whose deregulation is more robustly supported by the RNA-seq data.

We then tested more in detail FUS association with specific circRNA precursors by CLIP analysis on differentiated N2a cells. Nuclear extracts were subjected to FUS immunoprecipitation (Fig. 4a) and the recovered RNA was analysed with primers spanning the exon–intron junctions of the exons involved in the back-splicing event (Fig. 4b, 5′ and 3′ primers). As control for each precursor transcript, primers were designed on exon–intron splice junctions of either upstream or downstream regions at a distance of at least 10 kb (Fig. 4b, NEG primers). The NEG primers identify transcript regions that were verified in the RNA-seq data not to be affected by FUS levels (Supplementary Fig. 4c). The region between exon 7 and intron 7 of FUS pre-mRNA, known to bind FUS[48,49], was used as a positive control, while the pre-mRNA of ATP5o was used as a negative control[34] (Fig. 4c). Notably, compared to controls, circRNAs responding to FUS depletion/overexpression exhibited enrichment for FUS binding in at least one of the two splice junctions involved in the back-splicing event (Fig. 4d). In conclusion, these data indicate that FUS interacts with the pre-mRNA regions controlling the biogenesis of specific circRNAs.

In order to finally prove the direct involvement of FUS in the back-splicing reaction, we cloned the exon regions of c-03 and c-87 together with ~1,500 nucleotides of flanking introns into the pcDNA 3.1+ expression vector (Fig. 5a). Transfection in N2a cells together with scramble and FUS siRNAs indicated that both constructs were able to produce the corresponding circRNAs and to respond to FUS as the host endogenous transcripts (Fig. 5b and Supplementary Fig. 5). In particular, c-HA03 was downregulated while c-HA87 was upregulated upon FUS depletion. These data indicate that ~1,500 nucleotides are sufficient to drive circularization and to confer FUS dependence of the back-splicing reaction.

## Discussion

The discovery that circRNAs are widely expressed and highly conserved in all cell types has increased the potential impact of non-coding RNAs on cell function, adding to the complexity of regulatory processes. Moreover, in the nervous system where alternative splicing occurs more frequently than in other tissues, back-splicing further enlarges the number of transcript isoforms deriving from a primary transcriptional unit. It is now clearly established that circRNAs are not the result of splicing errors but instead, their biogenesis depends on the concerted activity of cis- and trans-acting factors.

In this study, we identified the RNA-binding protein FUS as a new regulator of circRNA production and we defined its

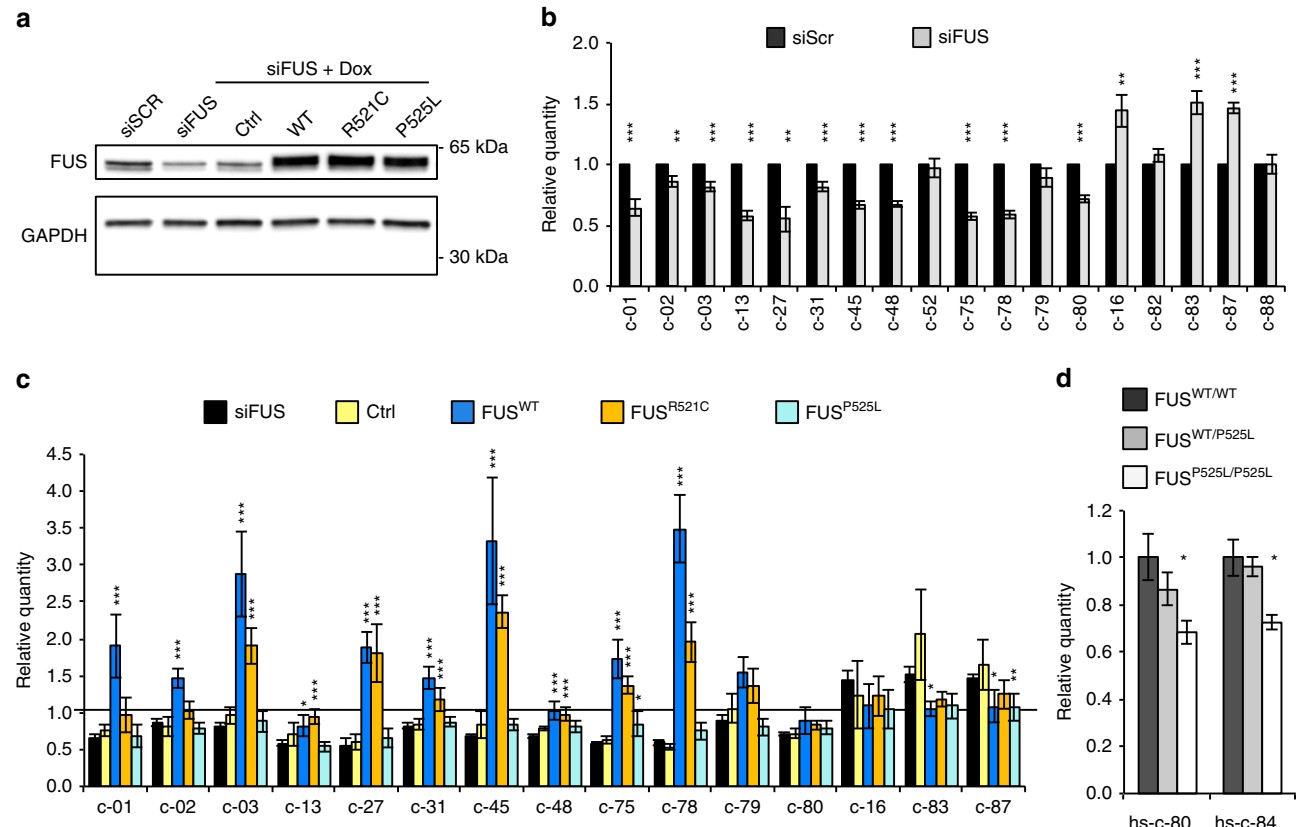

**Figure 3 | CircRNA expression upon ectopic expression of wild-type and mutant FUS.** (**a**) Western blot analysis on total protein extracts from differentiated N2a cells treated with control siRNAs (siScr) or with siRNAs against FUS (siFUS). In the same blot, shown also are proteins derived from differentiated N2a cells stably transfected with an Ctrl or with FUS[WT] (WT), FUS[R521C] (R521C) and FUS[P525L] (P525L) cDNA expression cassettes and treated with siFUS and Dox. GAPDH was used as a loading control. (**b**) Histograms show the levels of circRNAs in differentiated N2a cells treated with control siRNAs (siScr; black bars) or with siRNAs against FUS (siFUS; grey bars). CircRNA were quantified by qRT–PCR, normalized against Atp5o mRNA levels and expressed as relative quantity with respect to siScr samples set to a value of 1. (**c**) Histograms show the levels of circRNAs in the samples treated as in **a**; the values were normalized against Atp5o mRNA levels and siScr samples set to a value of 1. Statistical analysis was performed on Ctrl, FUS[WT], FUS[R521C] and FUS[P525L] samples against siFUS samples. (**d**) Histograms show the level of hsc-80 and hsc-84 in FUS[WT/WT], FUS[WT/P525L] and FUS[P525L/P525L] iPSC-derived MNs. CircRNAs were normalized against Atp5o mRNA levels and expressed as relative quantity with respect to FUS [WT/WT] samples set to a value of 1. For all the experiments shown in the figure, error bars represent s.e.m. of at least three independent experiments. *$P < 0.05$, **$P < 0.01$ and ***$P < 0.001$ correspond to two-tailed Student's t-test.

important role in controlling the expression of these molecules in mouse MNs. Considering its role in splicing regulation and in neurodegenerative disorders such as ALS and frontotemporal dementia, the study of FUS in circRNA biogenesis is important not only to elucidate its role in this process but also to link circRNA function to neurodegenerative processes.

The conditions of FUS depletion, even if not directly correlating with human mutations, represent a very suitable model system for studying nuclear loss of function of FUS and its impact on splicing processes.

Our data indicate that in vitro-derived MNs express a high number of circRNAs and that a specific subclass is affected by FUS levels. Interestingly, the observed variations in circRNA abundance were attributed to post-transcriptional events and not to transcriptional ones since, upon FUS depletion, the host linear transcripts did not vary or, in a minority of cases, changed in an opposite way with respect to the circRNA.

A more detailed analysis of a subgroup of circRNAs, validated by quantitative RT–PCR, indicated that FUS may either enhance or repress the back-splicing reaction. This is in line with previous studies indicating that FUS can act both as an activator and a repressor of splicing[50,51]. Notably, analysis of

subcellular localization led us to identify nuclear circRNA species entirely derived from exonic sequences. Until now, only intron-containing circRNAs were known to be localized in the nucleus[13]; therefore, these species may represent very interesting material for studying the function of circRNAs in this compartment.

Interestingly, our data demonstrate that circRNAs that decrease in $FUS^{-/-}$ cells also consistently diminish in conditions of FUS RNAi and inversely increase when FUS is ectopically expressed, indicating a direct relationship between the amount of back-splicing and the levels of FUS. CLIP experiments showed that FUS associates with intron regions proximal to the splice junctions involved in circRNA formation. This is in agreement with previous work indicating that FUS binds preferentially very long introns with preference for the 5′ end of the intron[47,51] and, in particular, around the alternatively spliced exons of genes associated with neuronal functions and neurodegeneration[50].

The use of the human $FUS^{R521C}$ and $FUS^{P525L}$ mutants, characterized by weak and strong cytoplasmic delocalization, respectively, has allowed to integrate the knockdown conditions (loss of function) with putative toxic gain-of-function activities due to specific mutations[52]. Interestingly, ALS-causative mutations have been shown to alter the splicing mode in

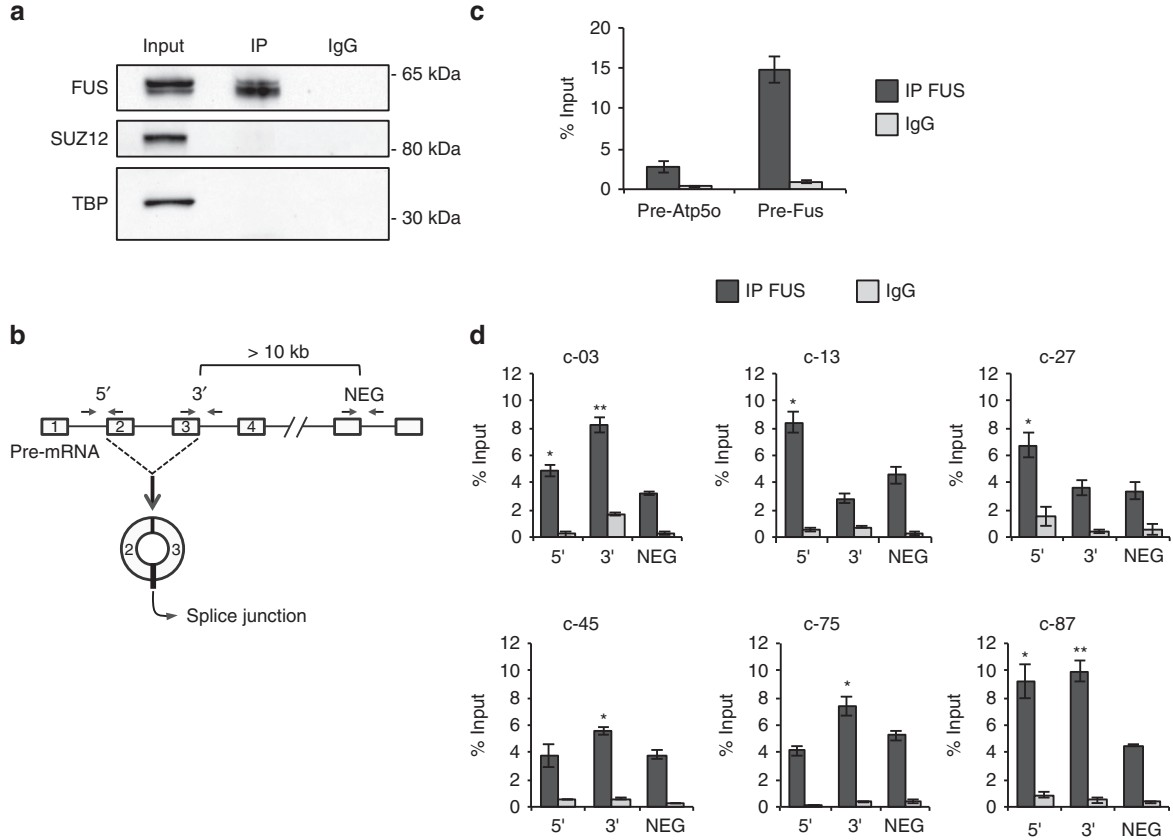

**Figure 4 | FUS binds the introns flanking the circularized exons.** (**a**) Protein extracts from FUS CLIP experiments, performed in differentiated N2a cells, were analysed by western blot with FUS antibodies and with SUZ12 and TBP antibodies as negative controls. Input samples account to 4% of the extract; IP and IgG represent 1/5 of the sample used for subsequent RNA analysis. (**b**) Schematic representation of the primers used for amplifying: the intron–exon junction upstream to the first exon included in the circRNA (5′); the exon–intron junction downstream to the last exon included in the circRNA (3′); the exon–intron junction at a distance of at least 10 Kb away from circularizing exons (NEG). (**c**) Histogram show the levels of enrichment in IP FUS and IgG samples for the Atp5o pre-mRNA (negative control—pre-Atp5o) and for the region across exon 7 and intron 7 of the FUS pre-mRNA (positive control—pre-Fus). The values are measured as a percentage of the input. (**d**) Histograms show the levels of enrichment of 5′, 3′ and NEG regions in IP FUS and IgG samples for six circRNA primary transcripts. The values are measured as a percentage of the input. For **c**,**d**, error bars represent s.e.m. of at least three independent experiments. *$P < 0.05$, **$P < 0.01$ and ***$P < 0.001$ corresponds to two-tailed Student's $t$-test.

a positive and negative manner through the 'gain' and 'loss' of interactions with specific splicing factors[31,46]. Likewise, the effects of 'gain' and 'loss' properties of mutant FUS are expected to be further complicated by their time-dependent progressive cytosolic accumulation observed in patients.

When considering the role of FUS on back-splicing, it is important to note that both decreased nuclear levels and altered interactions of the mutant proteins may affect the utilization of target splice junctions, thus altering the splicing mode of specific exon–intron domains of a primary transcript. Since FUS could also control the splicing of the host transcripts, a complex interplay between linear and back-splicing might be required to ensure the correct transcriptional outcome of specific genomic loci.

Notably, most of the circRNA species altered upon FUS depletion in murine MNs are conserved in human iPSCs-derived MNs, indicating that the activities of this novel class of transcripts may also be conserved. Two circRNAs appeared to be deregulated in a FUS-dependent manner similarly to the mouse, more directly linking circRNA production to the ALS pathology. The fact that this effect was evident only in the $FUS^{P525L}$ homozygous context is likely due to the experimental system used which is unable to reproduce the long-term features of ALS pathogenesis.

In conclusion, our study demonstrates that FUS regulates the splicing of a novel and abundant class of transcripts whose functions are still largely unknown. Moreover, we provide a novel system to study the role of circRNA-dependent regulatory networks in MNs, which may lead to new insights into the mechanism of mutant FUS-associated ALS and related disorders.

## Methods

**Oligonucleotides.** Oligonucleotide sequences used in this study are listed in Supplementary Table 1.

**Plasmid construction.** Plasmids for circRNA expression were constructed using pcDNA3.1 + backbone. To generate pc-HA03 and pc-HA87 constructs, two regions spanning intron 1 (∼1,500 nt)–exon 2 and exon 3–intron 2 (∼1,500 nt) of each corresponding host gene were PCR amplified from N2a genomic DNA using oligonucleotides listed in Supplementary Table 1. These two regions together with pcDNA3.1 + vector digested with HindIII-EcoRI for c-87, and with BamhI-XhoI for c-03, were ligated together using In-Fusion HD Cloning Kit (Clontech) according to the manufacturer's instructions. The HA tag was then inserted by inverted PCR using oligonucleotides listed in Supplementary Table 1.

**Cell cultures and treatments.** mESCs were cultured and differentiated as described in Wichterle *et al.*[40]. Briefly, generation of embryoid bodies (EBs) was achieved by culturing mESCs in ADNFK medium (1:1 Advanced DMEM/F12:Neurobasal medium, 10% Knock Out Serum Replacement (Gibco, 10828028),

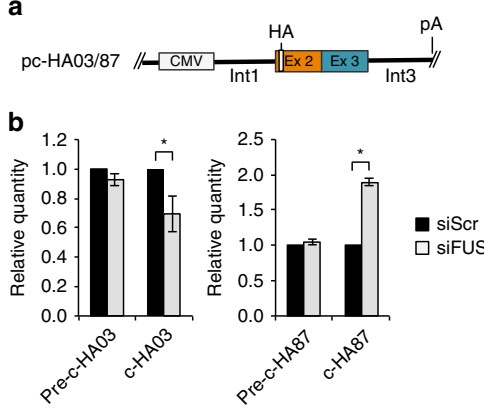

**Figure 5 | Artificial constructs reproduce FUS-dependent back-splicing.**
(**a**) Schematic representation of the pc-HA-c03 and pc-HA-c87 constructs. The second and third exons were cloned into pc-DNA3.1+ vector together with ~1,500 nucleotides of the flanking introns. An HA tag was inserted in order to discriminate the ectopically expressed circRNAs from the endogenous ones. (**b**) Histograms show the circRNA levels derived from pc-HA03 and pc-HA87 together with the corresponding linear transcripts (pre-c-HA03 and pre-c-HA87) in N2a cells treated with control siRNAs (siScr; black bars) or with siRNAs against FUS (siFUS; grey bars). The values were normalized against Neomycin mRNA levels and expressed as relative quantity with respect to siScr samples set to a value of 1. For all the experiments shown in the figure, error bars represent s.e.m. of three independent experiments. *$P < 0.05$ corresponds to two-tailed Student's t-test.

200 mM L-Glutamine, 0.1 mM 2-mercaptoethanol, 1× Pen/Strep). On day 2, ADNFK medium was complemented with 1× B27 Supplement (Gibco, 17504-044), 1 μM RA (Sigma Aldrich, R2625) and 0.5 μM SAG (Merck Millipore, 566660). EBs were expanded for 2 more days in these conditions and then disrupted by Papain dissociation system (Worthington Biochemical Corporation) following the manufacturer's instructions.

A specific reporter expressing myristoylated GFP under the control of the MNs promoter Hb9 was stably integrated in the mESCs derived from either $FUS^{+/+}$ or $FUS^{-/-}$ mice as described in Wichterle et al.[40]. Mature EBs were disrupted by Papain dissociation system (Worthington Biochemical Corporation) following the manufacturer's instructions. Hb9-GFP expressing cells were sorted in PBS using MoFlo Astrios cell sorter (Beckman Counter) configured with a 100 μm ceramic nozzle and operating at 28 psi. Purified GFP+ cells were collected by centrifugation and RNA was extracted by Direct-zol RNA MiniPrep (Zymo Research).

Culturing and differentiation of human iPSCs were performed as described in Lenzi et al.[34] with modifications. Briefly, human iPSCs were maintained in Nutristem-XF (Biological Industries) in plates coated with hESC-qualified matrigel (BD Biosciences) and passaged every 4–5 days with 1 mg ml$^{-1}$ dispase (Gibco). MNs were derived based on a previously described protocol modified to use adherent cells[53] as follows. Cells were cultured until confluent. Then the media was changed to neuron differentiation media: DMEM/F12:neurobasal 1:1 (Life Technologies) supplemented with nonessential amino acids, glutamax, B27 and N2 (Life Technologies). From day 0 to day 6, neuron differentiation media was supplemented with 1 μM RA (Sigma Aldrich), 1 μM smoothened agonist (Calbiochem), 0.1 μM LDN-193189 (Miltenyi Biotec) and 10 μM SB-431542 (Miltenyi Biotec). From day 7 to day 14, neuron differentiation media was supplemented with 1 μM RA, 1 μM smoothened agonist, 4 μM SU-5402 (Sigma Aldrich) and 5 μM DAPT (Sigma Aldrich).

A specific reporter expressing myristoylated GFP under the control of the MN promoter HB9 was stably integrated in the $FUS^{wt/wt}$, $FUS^{P525L/P525L}$ and $FUS^{WT/P525L}$ iPSCs cell line as described in Wainger et al.[41]. Briefly, this reporter system is based on a $HB9::GFP$ cassette embedded in a donor plasmid containing homology arms specific for the AAVS1 locus and a Puromycin resistance gene. This plasmid was co-transfected in iPSCs with AAVS1 ZFN (Sigma) with the Neon Transfection System (Life Technologies) as described in Lenzi et al.[34], and stable transfectants were isolated by Puromycin selection. Upon cell differentiation, iPSC-derived human MNs were dissociated and single-cell suspension were prepared for isolation. MNs were sorted based on GFP expression using a FACSAriaIII (Becton Dickinson, BD Biosciences) configured with a 100 μm

ceramic nozzle and operating at 19.84 psi. FACSAriaIII is equipped with a 488 nm laser and with the FACSDiva software (BD Biosciences version 6.1.3). Data were analysed using the FlowJo software (Tree Star). Following isolation, an aliquot of each tube of the sorted cells was evaluated for purity at the same sorter resulting in an enrichment >98–99% for each sample. Purified GFP+ cells were collected by centrifugation and plated on laminin-coated plates. After 7 days post-sorting, RNA was extracted by Direct-zol RNA MiniPrep (Zymo Research).

N2a cells, from ATCC (Cat. No. CCL-131), were cultured in DMEM medium D6546 (Sigma-Aldrich) supplemented with 10% fetal bovine serum (F7524, Sigma-Aldrich), L-glutamine (G7513, Sigma-Aldrich) and Penicillin–Streptomycin (P0781, Sigma-Aldrich). Differentiation was induced on 70% confluent plates with serum deprivation to a final concentration of 2% and 20 μM RA (R2625, Sigma-Aldrich).

SiRNA against FUS (5′-GAGTGGAGGTTATGGTCAA-3′)[50] and siScr (AllStars Neg. Control siRNA, 1027281, Qiagen) were transfected using Lipofectamine RNAiMAX Reagent (Thermo Fisher Scientific) according to the manufacturer's instructions.

For the generation of stable N2a cells expressing human FUS protein, the Flag-FUS$^{WT}$, Flag-FUS$^{R521C}$ and Flag-FUS$^{P525L}$epB-Puro-TT-derived plasmids[45] and epiggyBac transposase vector were transfected using Lipofectamine and Plus Reagent (Thermo Fisher Scientific). After 72 h, the cells were selected with Puromycin (1 μg ml$^{-1}$) treatment and the expression of the different forms of FUS protein was induced by adding Dox (0.2 μg ml$^{-1}$) to the culture medium 48 h before collecting the cells.

Nuclear/cytoplasmic fractionation was carried out by using the PARIS Kit (Ambion) on differentiated N2a cells following the manufacturer's specifications.

**Bioinformatic analysis.** Total RNA was extracted from sorted $Hb9::GFP+$ $FUS^{+/+}$ and $FUS^{-/-}$ cells and sequenced on a Illumina Hiseq 2,500 Sequencing system using TruSeq Stranded Total RNA Library Prep Kit with Ribo-Zero treatment (Illumina). An average of about 47 million 100 base pairs long paired-end reads were produced for each sample.

RNA-Seq reads were initially trimmed using the Trimmomatic software[54] to remove adapter sequences and poor quality bases. After that, Bowtie 2 (ref. 55) was used to align reads to a sequence database composed of rRNA, tRNA, snRNA, snoRNA and other non-coding species, which resulted to be overrepresented according to the FastQC software[56]; reads mapping linearly to these sequences were filtered out. The remaining reads were used for circRNA detection as follows: the two mates of each pair of reads (R1 and R2) were aligned separately to the reference genome (Bowtie 2 to GRCm38) and those mapping were discarded; the rest was used as input for find_circ[10]. First, 20 nt long anchors were produced from the ends of each read, then anchors were aligned to the reference genome with Bowtie 2 and alignments of both R1- and R2-derived anchors were used by find_circ to identify and count circular and linear splicing events, restricted by the presence of a GU/AG signal. Circular splicing events were then filtered for each biological sample separately for a few parameters, including: at least two unique reads mapping to head-to-tail junctions, distance between mapped anchors <100 kb, and mapping quality of the anchors of at least 35. Filtered circRNAs from each sample were then merged and, for all of them, reads mapping linearly to each of the two coordinates of the head-to-tail splice junction were parsed from the find_circ output and counted. We further excluded those putative back-splicing events that were spanning two non-overlapping genes, which were likely to be due to mapping errors.

To evaluate the differential expression of circRNAs between $FUS^{+/+}$ and $FUS^{-/-}$ conditions, we provided the edgeR software[57] with the read counts of both the back-splicing events and their cognate linear splicing events; the read counts of cognate linear splicing events were calculated summing all the reads mapping linearly on both the splice junctions involved in back-splicing. Events not having two or more reads in at least three samples were not tested for differential expression. Model fitting and testing was performed using the glmFIT and glmLRT functions. Given the low number of reads used for testing, we decided to use $P$ value instead of false discovery rate to select for differentially expressed events, setting the significance threshold value to 0.05.

The BEDTools software suite[58] was employed to annotate circRNAs by intersecting their genomic coordinates with those of the genomic features described in the Ensembl 77 gene annotation[59].

The comparison of the observed and the expected localization of circRNAs within the body of protein-coding transcripts was done as follows: circRNAs whose back-splicing junctions fall in an intron or outside the gene were filtered out; we then computed the distribution of the number of exons included between each circRNA back-splicing junction pair (for those cases in which the exonic structure of circRNAs could not be defined unambiguously due to the alternative splicing of the gene, we used the average number of exons rounded to the nearest integer); from the transcripts hosting these circRNAs, we randomly picked 5,000 groups of consecutive internal exons (used to simulate faux circRNAs), the number of exons of each group being sampled from the distribution of the number of exons previously computed; the localization of real and faux circRNAs with respect to untranslated regions and coding regions was determined; a chi-squared test was performed to determine whether real circRNAs show a preferential localization when compared to random faux circRNAs.

Raw reads from FUS HITS-CLIP experiment conducted by Lagier-Tourenne et al.[47] on whole mouse brain were downloaded from Gene Epression Omnibus. We identified FUS CLIP-Seq peaks for each of the three biological replicates separately. First, adapter and quality trimming of reads was performed using Trimmomatic; Cutadapt[60] was then used to remove all the adapter sequences that were not trimmed in the first phase. Trimmed reads were aligned to GRCm38 using Bowtie[61] with parameters -a -m 1 --best --strata. Duplicate reads, which could represent PCR artifacts, were removed using MarkDuplicates from Picard (picard.sourceforge.net/command-line-overview.shtml). Tools from the Pyicoteo[62] suite were used to call CLIP-Seq peaks. First, all reads were extended to a length of 36 nucleotides using the pyicos extend tool. Then, CLIP-Seq peak calling was performed using the pyiclip tool. Ensembl 77 GTF file was supplied to generate exploratory regions, using the option --region-magic genebody. For each replicate, only peaks with $P$ value < 0.0001 were retained. Finally, BEDTools merge with option -d 50 was used to merge peaks from all the replicates; all those peaks that resulted from the merge of peaks identified in at least two replicates were taken as FUS-binding sites. The two circRNAs whose deregulation parallels that of the linear counterpart were excluded from the analysis. The list of robustly expressed circRNAs was further reduced by removing all those circRNAs with back-splicing sites that did not coincide with annotated splice sites; this way, 1,026 out of the 1,151 circRNAs were retained. We also filtered out all the circRNAs that are flanked by introns shorter than 1,500 nt, narrowing the list down to 938 circRNAs. BEDTools intersect was used to find all the FUS-binding sites falling in the intronic regions flanking circRNA back-splicing sites.

**Protein extraction and western blot.** Whole-cell protein extracts were prepared from N2a cells using RIPA buffer and subjected to western blot analysis performed using precasted NuPAGE SDS-PAGE gels and reagents from Invitrogen. The immunoblots were incubated with the following antibodies diluted in 5% skim milk in TBS-T: FUS/TLS (sc-47711, Santa Cruz, 1:2,000), GAPDH (sc-32233, Santa Cruz, 1:2,000), SUZ12 (SC-46264, Santa Cruz, 1:200), and TBP (sc-273, Santa Cruz, 1:200). All the images were captured using the Molecular Imager ChemiDoc XRS+ (Bio-Rad), and the densitometric analyses were performed using the associated Image Lab software (Bio-Rad). Full scan of western blots are presented in Supplementary Fig. 7.

**RNA preparation and analysis.** Total RNA and RNA from nuclear/cytoplasmic fractionations was isolated using the Direct-zol RNA MiniPrep Kit with on-column DNAse treatment, according to the manufacturer's instructions (Zymo Research).

RNaseR treatment was performed on total RNA extracted from mouse EBs at day 6 of differentiation; 6 U of RNaseR (RNR07250, Epicentre) was used for 1 μg of RNA and the reaction was carried out for 15′ at 37 °C; the RNA was then extracted using the Direct-zol RNA MiniPrep Kit (Zymo Research).

For RNA retro-transcription, the SuperScript VILO cDNA Synthesis Kit was used (Thermo Fisher Scientific).

For the detection of circRNAs and their linear counterparts, cDNA samples were analysed by quantitative real-time PCR using PowerUp SYBR Green Master Mix (Thermo Fisher Scientific).

For the RNA extracted in CLIP experiments, semi-quantitative PCR was performed on cDNAs using MyTaq Red DNA Polymerase (Bioline) according to the manufacturer's instructions. The samples were then loaded on a 2.5% agarose gel. All the images were captured using the Molecular Imager ChemiDoc XRS+ (Bio-Rad), and the densitometric analyses were performed using the associated Image Lab software (Bio-Rad).

The oligonucleotides used in all the amplification steps are listed in Supplementary Table 1.

**Northern blot.** In all, 10 μg of total RNA from mESCs, $GFP^-$ $FUS^{+/+}$, $GFP^+FUS^{+/+}$ and was $GFP^+FUS^{-/-}$ were denatured with one volume of glyoxal loading dye (Ambion) for 30′ at 50 °C and loaded on 1.2% agarose gel. Electrophoresis was carried out for 2 h at 60 V. RNA was transferred on Hybond N+ membrane (GE Healthcare) by capillarity overnight in 10× SSC. Transferred RNA was cross-linked with UV at $1,200 \times 100 \mu J \, cm^{-2}$ and the membrane was washed in 50 mM Tris pH 8.0 for 20 min at 45 °C. Prehybridization and hybridization were performed in NorthernMax buffer (Ambion) at 68 °C (30 min and overnight, respectively). A total of 500 ng of DIG-labelled probe in 10 ml were used for hybridization. The membrane was then washed with 2× SSC 0.1% SDS twice 30 min, then once 30 min and once 1 h with 0.2× SSC 0.1% SDS at hybridization temperature. The membrane was the processed for DIG detection (hybridization with anti-DIG antibody, washing and luminescence detection) with the DIG Luminescence Detection Kit (Roche), according to the manufacturer's instructions. DIG-labelled probes were produced by in vitro transcription with DIG-RNA labelling mix (Roche) of PCR templates produced with the oligonucleotides listed in Supplementary Table 1. Transcription with T7 RNA polymerase (Promega) was carried out for 2 h and the RNA was then resolved and purified by electrophoresis on denaturing polyacrylamide gel. Full scan of northern blots are presented in Supplementary Fig. 6.

**Nuclear cross-linking and immunoprecipitation assay.** Plates ($10 \times 10 \, cm^2$) of 20% confluent differentiated N2a cells were UV cross-linked at $4,000 \times 100 \mu J \, cm^{-2}$ energy. Cells were resuspended in nuclear isolation buffer (the final concentration of the buffer is: 256 mM sucrose; 8 mM Tris-HCl pH 7.5; 4 mM MgCl$_2$; 0,8% Triton X-100) and immunoprecipitation was performed as previously described by Rinn et al.[63]. Cells lysate was sonicated with Bioruptor sonication device (Diagenode) and nuclear membrane and debris were pelleted by centrifugation at 13,000 r.p.m. for 10 min at 4 °C. Cell lysates were precleared with Protein G Agarose/Salmon Sperm DNA (16–201, Merck Millipore) before incubation with either FUS Antibody (sc-47711, Santa Cruz) or mouse IgG (sc-2025, Santa Cruz). Four washes were performed with RIP buffer (150 mM KCl, 25 mM Tris pH 7.4, 5 mM EDTA, 0.5 mM DTT, 0.5% NP40) and two washes with RIP High-Salt buffer (500 mM KCl, 25 mM Tris pH 7.4, 5 mM EDTA, 0.5 mM DTT, 0.5% NP40). Before RNA extraction, 1/5 of the cell lysate was used for western blot analysis. RNA fraction was treated with Proteinase K (AM2546, Thermo Fisher Scientific) at 45 °C for 50 min; the samples were then placed 10 min at 95 °C and finally the RNA was extracted using the miRNeasy Mini Kit with on-column DNAse treatment, according to the manufacturer's instructions (Qiagen).

**Data availability.** RNA sequencing raw data have been deposited at Gene Expression Omnibus (GSE83226).

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

## Acknowledgements

We thank M. Marchioni and T. Santini for technical support. We thank Professor K. Eggan for providing the AAVS1 Hb9::GFP system. This work was partially supported by grants from: ERC-2013 (AdG 340172–MUNCODD), AriSLA full grant 2014 'ARCI', Epigen-Epigenomics Flagship Project, Human Frontiers Science Program Award RGP0009/2014, AFM-Telethon (17835), Fondazione Roma and PRIN. This work was also partially supported by NIH/NINDS grant R01 NS07377 to N.A.S.; L.L. was supported by the Judith and Jean Pape Adams Foundation.

## Author contributions

L.E. and S.D.M. performed circRNA analysis in murine motor neurons, circRNAs analysis in human motor neurons, RNAi experiments and FUS overexpressing experiments. S.D.M. performed ClIP experiments. L.L., N.S., P.L. and D.C. performed mESC differentiation to motor neurons. D.C. analysed the markers of murine motor neuron differentiation. I.L. and A.C. performed circRNA expression bioinformatic analysis. A.C. performed statistical analysis on RNA-seq results and analysed public ClIP data. A.R. and R.D.S. performed human iPSCs differentiation to motor neurons and analysed the markers of human motor neuron differentiation. R.S. and L.E. performed RNAseR treatment and generated circRNA-expressing plasmids. R.S. and S.D.M. performed nucleus/cytoplasmic fractionations. M.M. performed northern blot analyses. G.P. performed flow cytometric analysis and cell sorting. E.C. contributed to writing the paper. M.M. and I.B. coordinated the research and wrote the paper.
