## [Peer Review File · Nature Communications]

Reviewers' comments:

Reviewer #1 (Remarks to the Author):

In the ms by Stefano et.al, the authors identified 1) >3000 circRNAs in ES-cell derived motor neurons and 2) FUS as an important trans-factor in regulating circRNA biogenesis in motor neurons. Given the important role of FUS in motor neuron degeneration and the mouse-human conservation of considerable circRNAs identified in motor neurons, these findings are of great relevance not only to the basic research on circRNA biology, but also to the study of circRNA in human diseases. I have a few comments/suggestions,

1. In the circRNA identification part, did the authors use only the known/annotated junctions or also predict the novel splicing junction? If the former is the case, how could they find circRNAs in the intergenic part (Figure 1 b) and junctions within introns (Figure 1C)? If the latter is true, the author should explain how the novel splicing junctions were defined?
2. For the readers better to understand Figure 1C, the authors may provide Genome browser plot for representative circRNAs from each category in a supplementary figure.
3. In 'CircRNA expression is modulated in FUS^{-/-} motor neurons' line 128 and line 141, the two sentences are inconsistent, while line 128 claimed that all 23 was verified, line 141 was clear that only 19 showed concordant results.
4. In the analysis of FUS binding sites, what the authors did has limited spatial resolution and was not in genome-wide manner, I would ask them to perform iCLIP or PAR-CLIP to study the FUS binding at global level and at nucleotide resolution. This will allow to better understand the relation between FUS binding and circRNA biogenesis.

Reviewer #2 (Remarks to the Author):

In the present study, Stephano et al. first investigated the expression profile of circular RNAs (circRNAs) in mouse embryonic stem cell-derived spinal motor neurons (MNs), and identified a class of circRNAs that are enriched in spinal MNs. The authors then demonstrated that the expression levels of 17 members of these MN-enriched circRNAs are affected in FUS knockout MNs, and 15 of the 17 circRNAs of interest were also expressed in human iPS-derived MNs. The involvement of FUS to regulate these circRNAs biogenesis was further validated in FUS knockdown and overexpressed N2a cells. The change of circRNA expression seems to be controlled at post transcriptional level as those corresponding linear host transcripts were unaffected or only slightly modulated upon FUS KD/OE. The authors further performed cross-linking immunoprecipitation to demonstrate that FUS binds to circularizing exon-intron junctions, and they suggest that FUS might participate in backsplicing of circRNA directly. In addition, the authors compared circRNA levels in wild-type FUS⁻ and two fALS-linked FUS mutants (R521C and P525L)-overexpressing N2a cells and found partial loss-of-functional effects of FUSR521C and FUSP525L on circRNA biogenesis, which might be attributed to the reduction of nuclear FUS levels.

Being a novel category of non-coding RNA, circRNAs have recently been shown to be conserved between species, and have emerged as new regulatory RNA involved in both

development and/or disease 1. This manuscript aims to identify MN enriched circRNAs and to uncover the involvement of FUS in circRNA biogenesis, thus providing elements of novelty. However, the current data at this stage, in my opinion, is still premature and lacks of several key experiments to consolidate their findings of MN-enriched circRNAs, FUS-dependent circRNA biogenesis mechanisms and the potential link to ALS. These undermine the significance and robustness of this study.

Specific comments are listed below:

1. In Fig2:

A. The authors mentioned, "by this selection process we validated 17 RNA species as circular molecules selectively expressed in bona fide motor neurons in vitro." Yet I found the current data is not sufficient enough to support this statement. The expression level of circRNAs of interest should also be quantified in ESCs, EBs (or FACS-sorted GFP-negative cells) together with FACS-sorted GFP-positive cells (they did some, but not all) to show the specificity of expression. Additionally, it would be more comprehensive if the change of linear transcripts shown in supplementary figure 4c and d are combined with quantified data from figure 2b to clearly show that the discordancy of the tendency of change of linear and circular transcripts. Similar concern is raised on the verification of circRNA expression in human iPS-derived motor neurons.

B. To give solid illustration of FUS-related change of selected circRNAs, the N numbers, SDs, and P values should be provided in fig2b and fig2c. Due to the potential bias and artifacts of RT-PCT readout, it would be more convincing to design additional primer sets to validate the change of identified circRNA expression in WT and FUS KO MNs, to verified by Northern Blotting.

C. Moreover, as identification of these MN enriched circRNAs is a key conclusion in this report, the authors should further show that their in vitro findings do recapitulate in vivo conditions, as the MN differentiation protocol only acquires cervical identity. For example, it would be nice to use situ hybridization to reveal the temporal expression of selected circRNAs in developing mouse spinal cord 2.

2. In Fig3:

A. Knockdown of FUS in N2a cells seems redundant. For overexpression experiments, as authors stated that at least for some circRNAs, baseline level of FUS is sufficient for controlling back-splicing, the overexpression of mutant FUS should be done on FUS KO background to eliminate the effect of endogenous FUS.

B. Importantly, the data presented from WT/mutant FUS overexpressing N2a cells is not sufficient to provide clinical correlation with ALS, and is unable to clarify whether circRNA generation is affected by nuclear level or enzymatic activity of FUS. For disease prospective, it would be more convincing to show that those identified circRNAs are also dysregulated in human MNs derived from ALS iPSs (FUSR521C and FUSR514S and FUSP525L) which have been established in the authors' previous work 3

In Fig4:

A. To provide a more comprehensive analyses of FUS-mediated back-splicing, CLIP-Seq for FUS should be performed to show specific and enriched binding of FUS to exon-intron junctions of circRNAs. Alternatively, the authors can analyse reported FUS-CLIP-seq results 3-6 to further verify the selective binding of FUS on back-splicing sites. If this is not the case, then how does FUS pervasively bind to exon-intron junctions and determine the linear/circular balance?

B. I also think that the evidence of FUS to regulate back splicing directly is missing. As FUS participates in several RNA metabolism pathways, including the FUS-miRNA autoregulatory loop identified from the same lab. Is it possible that miRNA might target to other proteins/RNAs involved in splicing machineries and/or circRNA biogenesis (Adar1, etc)?

Minor points:

3. The second and third paragraphs of Discussion are more like introduction. The authors should reorganise their discussions.

4. For statistic analysis, it is confusing that some of the results were analyzed by one-tailed Student's t-test (Fig 3c, Fig S4c,d). All of the results in this study should be consistently analyzed by two-tailed Student's t-test.

5. There is an additional label of $\text{Pdgr-}\alpha$ in Fig S1c.

References

1 Zheng, Q. et al. Circular RNA profiling reveals an abundant circHIPK3 that regulates cell growth by sponging multiple miRNAs. *Nature communications* 7, 11215, doi:10.1038/ncomms11215 (2016).

2 You, X. et al. Neural circular RNAs are derived from synaptic genes and regulated by development and plasticity. *Nature neuroscience* 18, 603-610, doi:10.1038/nn.3975 (2015).

3 Lenzi, J. et al. ALS mutant FUS proteins are recruited into stress granules in induced pluripotent stem cell-derived motoneurons. *Disease models & mechanisms* 8, 755-766, doi:10.1242/dmm.020099 (2015).

4 Lagier-Tourenne, C. et al. Divergent roles of ALS-linked proteins FUS/TLS and TDP-43 intersect in processing long pre-mRNAs. *Nature neuroscience* 15, 1488-1497, doi:10.1038/nn.3230 (2012).

5 Nakaya, T., Alexiou, P., Maragkakis, M., Chang, A. & Mourelatos, Z. FUS regulates genes coding for RNA-binding proteins in neurons by binding to their highly conserved introns. *Rna* 19, 498-509, doi:10.1261/rna.037804.112 (2013).

6 Rogelj, B. et al. Widespread binding of FUS along nascent RNA regulates alternative splicing in the brain. *Scientific reports* 2, 603, doi:10.1038/srep00603 (2012).

Reviewer #3 (Remarks to the Author):

Dini Modigliani & Errichelli et al. investigate the role the RNA binding protein FUS in the regulation of circular RNA expression in motoneurons. They first performed a genome wide identification of circRNAs expressed in mESC-derived motoneurons. Then they highlighted a sub-population of circRNA whose expression is affected by variation of FUS protein level. Finally, the authors used immortalized cell lines to explore a potential direct role of FUS in circRNAs biogenesis.

Identification of circRNAs in motoneurons was performed in a well-established and well-controlled cellular model; which thus could give some robust support for future studies in the field. Another strength of the study arises from the set of diverse experiments revealing

the impact of FUS levels on circRNA expression. Nevertheless, there are significant open questions regarding the role of FUS, leaving it unclear whether the protein indeed directly contributes to circRNA biogenesis or whether the apparent phenotypes arise indirectly through disruption of other cellular processes.

Considering the increasing number of publications providing unbiased identifications of circRNAs in various physiological models, it is essential that the authors test more directly a role for FUS in circRNAs biogenesis.

Major points

1. Characterization of circRNA

According to the RNAseq data, expression of the circRNAs examined is very low and quite variable between biological replicates. Many circRNAs are detected with only one or two reads. This raises concern regarding their existence and physiological relevance. While the authors performed extensive RT-PCR validations, this approach can often give rise to artifacts, in particular when starting material is low. Thus the authors should (at least for a few circRNA species) validate their existence by a more rigorous approach (northern blot or RNase protection assays).

2. Role of FUS in circRNA biogenesis

a. Much of the data in the manuscript is correlative, demonstrating alterations in circRNAs when FUS levels are elevated or decreased. Using Clip assays the authors attempt to probe direct association of FUS with segments that undergo circularization. However, these experiments do not strongly support a preferential association of FUS with introns proximal to circularization sites. Only a few candidates were tested and the observed enrichment of FUS binding in these sequences is very modest. The authors should perform a broader analysis by using genome wide analysis of FUS binding sites (some of this data is available) and assess the proportion of FUS-dependent circRNAs whose pre-mature transcripts bind FUS.

b. Impact of FUS protein level on circRNA expression could be explained by multiple reasons and the causality of altered circRNAs in FUS mutants is unclear. While the data on linear RNAs support that FUS action is independent of transcriptional regulation, other hypotheses cannot be ruled out (impact on circRNA stability, indirect role of FUS through overall modification of molecular repertoire in FUS^{-/-} cells...). To test this more rigorously, the authors could use a minigene reporter approach and assess more directly a role for FUS by mutating its binding site in the reporter.

3. Figure 2b/c: The expression level of the cognate linear RNAs should be displayed.

Minor points

1. The procedure of RNA sequencing and the data analysis needs to be more clearly described. For example:

- Read length?

- Criteria to consider a read spanning a back-splicing junction (number of nucleotide from each side of the junction)?
- How are the head-to-tail junctions defined (GU/AG dinucleotides)?
- What statistical analysis was used to define circRNAs differentially regulated between FUS+/+ and Fus-/- conditions?

2. The meaning of Figure 1b is unclear. The authors attempt to estimate the proportion of circRNAs hosted by coding versus non-coding genes. However, in the text they stated that reads coming from several non-coding genes were discarded in their analysis (line 92). Thus, the numbers in Figure 1b are likely to be misleading.

3. Figure 1d: Keeping the same scale for x and y-axis would help the readers to compare the fold-changes of the circRNAs and their cognate linear RNAs.

4. Figure 2: the number of independent biological replicates should be indicated in the figure legends.

5. The authors state that the nuclear circRNAs c-01, c-087 and c-088 are devoid of intron sequences (line 175). What are the supporting evidences of absence of intronic sequences? It appears that only junctions were analyzed, not the entire sequence of the circRNAs.

Here below the major changes and the new experiments performed.

- *Analysis of FUS CLIPseq data has been performed. An interesting correspondence was found: intronic regions flanking circRNAs deregulated by FUS KO are enriched in FUS binding sites compared to intron region flanking unaffected circRNAs.*
- *Quantitative analyses of circRNA levels were performed in the different conditions requested. The results confirm previous semi-quantitative data and strengthen the conclusions.*
- *Northern blots were performed for some circRNAs confirming qRT-PCR data.*
- *Following the request, we performed the rescue experiments with the WT and mutants FUS proteins in FUS KD conditions.*
- *Comparative analysis in human was extended to iPS-derived motor neurons carrying the FUS^{P525L} mutation. Two species were identified that respond to FUS alterations similarly to the murine counterparts.*
- *Artificial constructs containing the exons and part of the flanking intron regions involved in back-splicing event of two different circRNAs were produced. They showed ability to promote circularization in a FUS-dependent manner similarly to the endogenous species.*

Reply to reviewers' comments:

Reviewer #1

In the ms by Stefano et.al, the authors identified 1) >3000 circRNAs in ES-cell derived motor neurons and 2) FUS as an important trans-factor in regulating circRNA biogenesis in motor neurons. Given the important role of FUS in motor neuron degeneration and the mouse-human conservation of considerable circRNAs identified in motor neurons, these findings are of great relevance not only to the basic research on circRNA biology, but also to the study of circRNA in human diseases. I have a few comments/suggestions,

1. In the circRNA identification part, did the authors use only the known/annotated junctions or also predict the novel splicing junction? If the former is the case, how could they find circRNAs in the intergenic part (Figure 1 b) and junctions within introns (Figure 1C)? If the latter is true, the author should explain how the novel splicing junctions were defined?

The procedure does not involve the use of a reference transcriptome: reads are first segmented, then aligned directly to the genome to perform a spliced alignment (see details in Methods). In this way, already known and novel back splice junctions can be identified. This has been clarified in the Results section.

2. For the readers better to understand Figure 1C, the authors may provide Genome browser plot for representative circRNAs from each category in a supplementary figure.

We added Genome browser plots in Supplementary Fig. 1d.

3. In 'CircRNA expression is modulated in FUS^{-/-} motor neurons' line 128 and line 141, the two sentences are inconsistent, while line 128 claimed that all 23 was verified, line 141 was clear that only 19 showed concordant results.

We agree that there was some intricate numerology (21 were the species whose circularity was checked for RNaseR resistance and only 19 passed this test). We have made it clearer. The referee can also value that (as requested by another reviewer) we substituted RT-PCR with quantitative real time analysis. The qRT-PCR confirmed previous data. In conclusion, we now show 19 species which passed the RNaseR screening and which show FUS-dependent accumulation (see new figure 2 and Supplementary Fig.2b).

4. In the analysis of FUS binding sites, what the authors did has limited spatial resolution and was not in genome-wide manner, I would ask them to perform iCLIP or PAR-CLIP to study the FUS

binding at global level and at nucleotide resolution. This will allow to better understand the relation between FUS binding and circRNA biogenesis.

We analysed public FUS CLIP data (Lagier-Tourenne et al., 2012 Nat. Neurosci.) and found that intronic regions flanking circRNAs deregulated by FUS KO are enriched in FUS binding sites compared to those flanking unaffected circRNAs. The results of this genome wide analysis are now described in the new Supplementary Fig.4a and b.

Reviewer #2 (Remarks to the Author):

In the present study, Stephano et al. first investigated the expression profile of circular RNAs (circRNAs) in mouse embryonic stem cell-derived spinal motor neurons (MNs), and identified a class of circRNAs that are enriched in spinal MNs. The authors then demonstrated that the expression levels of 17 members of these MN-enriched circRNAs are affected in FUS knockout MNs, and 15 of the 17 circRNAs of interest were also expressed in human iPS-derived MNs. The involvement of FUS to regulate these circRNAs biogenesis was further validated in FUS knockdown and overexpressed N2a cells. The change of circRNA expression seems to be controlled at post transcriptional level as those corresponding linear host transcripts were unaffected or only slightly modulated upon FUS KD/OE. The authors further performed cross-linking immunoprecipitation to demonstrate that FUS binds to circularizing exon-intron junctions, and they suggest that FUS might participate in backsplicing of circRNA directly. In addition, the authors compared circRNA levels in wild-type FUS- and two fALS- linked FUS mutants (R521C and P525L)-overexpressing N2a cells and found partial loss-of-functional effects of FUSR521C and FUSP525L on circRNA biogenesis, which might be attributed to the reduction of nuclear FUS elevels.

Being a novel category of non-coding RNA, circRNAs have recently been shown to be conserved between species, and have emerged as new regulatory RNA involved in both development and/or disease 1. This manuscript aims to identify MN enriched circRNAs and to uncover the involvement of FUS in circRNA biogenesis, thus providing elements of novelty. However, the current data at this stage, in my opinion, is still premature and lacks of several key experiments to consolidate their findings of MN-enriched circRNAs, FUS-dependent circRNA biogenesis mechanisms and the potential link to ALS. These undermine the significance and robustness of this study.

Specific comments are listed below:

1. In Fig2:

A. The authors mentioned, "by this selection process we validated 17 RNA species as circular molecules selectively expressed in bona fide motor neurons in vitro." Yet I found the current data is not sufficient enough to support this statement. The expression level of circRNAs of interest should also be quantified in ESCs, EBs (or FACS-sorted GFP-negative cells) together with FACS-sorted GFP-positive cells (they did some, but not all) to show the specificity of expression.

The term "selectively" was utilized by mistake in that specific sentence; in fact, in previous Fig.2a we showed that some circRNAs were already expressed in ES cells while others were enriched in MNs. As requested, for all circRNAs we repeated the experiments by performing qRT-PCR analysis on triplicates of ES cells in parallel with FACS-sorted GFP-negative versus GFP-positive cells and identified nine circRNAs which can be now defined as "enriched in motor neurons". The new data, included in Fig. 2b, are shown together with the qRT-PCR analysis in FUS^{+/+} versus FUS^{-/-} GFP-positive cells (Fig. 2a). The data obtained are in agreement with the previous semi-quantitative analysis.

Additionally, it would be more comprehensive if the change of linear transcripts shown in supplementary figure 4c and d are combined with quantified data from figure 2b to clearly show that the discordancy of the tendency of change of linear and circular transcripts. Similar concern is raised on the verification of circRNA expression in human iPS-derived motor neurons.

In the previous version of the paper, the data relating to the abundance of the linear transcripts derived solely from RNAseq data. Instead, those shown in Supplementary Figure 4c and d referred to the experiments performed in N2A cells. Therefore, as requested, we performed qRT-PCR analysis of the linear

forms in GFP⁺-FUS^{+/+} and GFP⁺-FUS^{-/-} cells. The data are shown in the new Supplementary Fig. 2e and f. The results confirm all previous data and show that, despite circRNA modulation, there is no significant effect on the linear counterpart, therefore indicating that the effects observed are not due to transcriptional control. The linear counterparts in human were not analyzed since there was no comparison between FUS^{+/+} and FUS^{-/-} conditions.

B. To give solid illustration of FUS-related change of selected circRNAs, the N numbers, SDs, and P values should be provided in fig2b and fig2c. Due to the potential bias and artifacts of RT-PCT readout, it would be more convincing to design additional primer sets to validate the change of identified circRNA expression in WT and FUS KO MNs, to verified by Northern Blotting.

We performed qRT-PCR on RNA from three independent experiments and the values with the statistics are now shown in the histograms in fig 2a and b. We also performed Northern blot analyses for the two most expressed circRNAs (c-78 and c-31). In the new panel of Fig 2c we show the abundance of these species in FUS^{+/+} ES cells in parallel with FACS-sorted GFP-negative versus GFP-positive FUS^{+/+} cells and FUS^{-/-} MNs.

C. Moreover, as identification of these MN enriched circRNAs is a key conclusion in this report, the authors should further show that their in vitro findings do recapitulate in vivo conditions, as the MN differentiation protocol only acquires cervical identity. For example, it would be nice to use situ hybridization to reveal the temporal expression of selected circRNAs in developing mouse spinal cord.

The procedure we used to differentiate motor neurons from mouse embryonic stem cells was established in Wichterle's lab (Wichterle and Peljto, 2008) through a protocol which recapitulates in vitro the patterning signals relevant during motor neuron physiological development (Retinoic Acid pathway and Sonic Hedgehog pathway). We have better specified in the text the MN markers analyzed and reported in Supplementary Fig.1c.

In particular, the paper states: "...efficient induction of spinal neural identity is achieved when embryoid bodies are treated with Retinoic Acid 2 days after the onset of differentiation, at a stage when cells acquire characteristics of primitive ectoderm. One day after the addition of Retinoic Acid, cells acquire early neural identity and can be patterned with Hedgehog to induce expression of ventral neural markers and specify motor neuron progenitor identity...." (Wichterle and Peljto, 2008).

The application of this procedure allowed us to obtain a population of embryoid bodies highly enriched in spinal motoneurons further isolated by the expression of a motoneuron-specific GFP reporter gene. Moreover, in GFP-positive cells we observed the specific upregulation of ChAT and Islet-1, markers of MN differentiation.

We agree that the use of in situ hybridization technique to visualise circRNAs would be a nice approach to address the specific and temporal expression of these molecules mouse spinal cord. Indeed, it represents a developing area of research in the lab even though the conditions to specifically distinguish the circular forms the linear counterpart still need to be set up.

2. In Fig3:

A. Knockdown of FUS in N2a cells seems redundant. For overexpression experiments, as authors stated that at least for some circRNAs, baseline level of FUS is sufficient for controlling back-splicing, the overexpression of mutant FUS should be done on FUS KO background to eliminate the effect of endogenous FUS.

We agree with the referee and we completely reset the experiment. Following the suggestion, we performed the overexpression experiment upon FUS KD with both the WT and mutants FUS proteins. The results are shown in the new Fig. 3c and indicate the modulation of circRNA biogenesis correlates with alteration of the nuclear levels of FUS as well as with putative toxic gain of function activities. Here below the results as presented in the revised version of the paper:

"With respect to a control cell line carrying an empty vector (Ctrl), the ectopic expression of FUS^{WT} was able to rescue the correct expression levels of almost all circRNAs (Fig. 3c). For those species which were down-regulated in FUS RNAi, FUS^{R521C} and FUS^{P525L} failed to fully rescue circRNA levels with the strongest

effect observed with FUS^{P525L} , the more mislocalized of the two mutant FUS proteins. Even if a simple hypothesis would correlate this phenotype with the amount of nuclear FUS , it cannot be excluded that the mutations per se lead to a loss of activity in splicing regulation; in fact, it was previously shown that both the FUS^{R521C} and FUS^{P525L} lead to decreased interactions with splicing promoting factors, such as the $U1-70K^{32}$. Therefore, by loosing such interaction the mutant proteins could affect the proper utilization of specific splice junctions more sensitive to $U1$ snRNP recognition.

For the circRNAs up-regulated upon FUS depletion, the FUS^{R521C} and FUS^{P525L} proteins were able to reduce circRNAs at the same levels as FUS^{WT} . Also in this case the results can be explained by two different models: either low levels of nuclear FUS are sufficient to inhibit circularization, or the $R521C$ and $P525L$ mutations confer stronger back-splicing repressive activity. Both mutants were indeed shown to have a stronger binding than the WT protein to splicing-related factors, such as the SMN complex, thus interfering with snRNP production and decreasing splicing efficiency^{32,47}.

B. Importantly, the data presented from WT /mutant FUS overexpressing N2a cells is not sufficient to provide clinical correlation with ALS, and is unable to clarify whether circRNA generation is affected by nuclear level or enzymatic activity of FUS . For disease prospective, it would be more convincing to show that those identified circRNAs are also dysregulated in human MNs derived from ALS iPSs ($FUSR521C$ and $FUSR514S$ and $FUSP525L$) which have been established in the authors' previous work .

We agree with the reviewer that the correlation with the disease is very important even if the focus of the paper at this stage was to analyze FUS behaviour in directing back-splicing reactions. However, we followed referee's suggestion and tested the variation of expression of our selected candidates on RNA from IPS-derived MNs carrying the $P525L$ FUS mutation in homozygous ($FUS^{P525L/P525L}$) and heterozygous ($FUS^{WT/P525L}$) conditions. We selected the $P525L$ mutation since is the one with the strongest delocalization phenotype. The results show that among the conserved species, two responded in human ALS mutant context similarly to the mouse but only in homozygous conditions. We have inserted these data in Fig. 3d. It is important to remind that the in vitro differentiation of patient-derived iPSCs cannot be expected to reproduce exactly the in vivo conditions since FUS accumulation in the cytoplasm (combined with possible toxic effects) is a continuous process that occurs in several decades. We previously showed that only in homozygous conditions it is possible to visualize clear alterations in differentiated iPSC (Lenzi et al., 2015). In fact, in heterozygous conditions the effects of the mutation are much weaker as shown by the lack of FUS autoregulation in pre-mRNA splicing; effect that was only detectable in $FUS^{P525L/P525L}$ conditions (Lenzi et al. 2015). Even if FUS KO does not correspond to any pathological genetic background, it is however relevant for providing a nuclear loss of function condition that in the pathology is reached only after very long periods of time.

In Fig4:

A. To provide a more comprehensive analyses of FUS -mediated back-splicing, CLIP-Seq for FUS should be performed to show specific and enriched binding of FUS to exon-intron junctions of circRNAs. Alternatively, the authors can analyse reported FUS -CLIP-seq results 3-6 to further verify the selective binding of FUS on back-splicing sites. If this is not the case, then how does FUS pervasively bind to exon-intron junctions and determine the linear/circular balance?

As indicated for referee#1, we analysed public FUS CLIP data (Lagier-Tourenne et al., 2012 Nat. Neurosci.) and found that intronic regions flanking circRNAs deregulated by FUS KO are enriched in FUS binding sites compared to those flanking unaffected circRNAs. The results of this genome wide analysis are now described in the new Supplementary Fig 4a and b.

B. I also think that the evidence of FUS to regulate back splicing directly is missing. As FUS participates in several RNA metabolism pathways, including the FUS -miRNA autoregulatory loop identified from the same lab. Is it possible that miRNA might target to other proteins/RNAs involved in splicing machineries and/or circRNA biogenesis (Adar1, etc)?

We cannot exclude that other protein factors (such as Adar1) may contribute directly or indirectly to the back-splicing events affected by FUS depletion. However, CLIP experiments allowed us to establish that, at least for 6 circRNAs, FUS is able to bind the surrounding intron sequences of the circularizing exons, suggesting its direct involvement in the biogenesis of these circRNAs. In order to strengthen this point, we raised specific constructs containing the circularizing exon plus ~1500 nucleotides of flanking introns and tested their ability to produce circRNA in the presence or absence of FUS. Fig.5 shows that two such clones (pc-HA03 and pc-HA87) were able to produce the corresponding circRNAs and that this activity was responsive to FUS. In particular, c-HA03 resulted down-regulated and c-HA87 up-regulated. These results confirm the involvement of FUS in directing the back-splicing reaction in both a positive and negative way and that ~1500 nucleotides are enough to provide such responsiveness in agreement with CLIP data.

Minor points:

3. The second and third paragraphs of Discussion are more like introduction. The authors should reorganise their discussions.

We agree; in the present version we moved a large part of the Discussion in the Introduction.

4. For statistic analysis, it is confusing that some of the results were analyzed by one-tailed Student's t-test (Fig 3c, Fig S4c,d). All of the results in this study should be consistently analyzed by two-tailed Student's t-test.

We changed the statistic analysis for all the experiments with two-tailed Student's t-test.

5. There is an additional label of Pdgfr- α in Fig S1c.

We fixed this.

Reviewer #3 (Remarks to the Author):

Dini Modigliani & Errichelli et al. investigate the role the RNA binding protein FUS in the regulation of circular RNA expression in motoneurons. They first performed a genome wide identification of circRNAs expressed in mESC-derived motoneurons. Then they highlighted a sub-population of circRNA whose expression is affected by variation of FUS protein level. Finally, the authors used immortalized cell lines to explore a potential direct role of FUS in circRNAs biogenesis.

Identification of circRNAs in motoneurons was performed in a well-established and well-controlled cellular model; which thus could give some robust support for future studies in the field. Another strength of the study arises from the set of diverse experiments revealing the impact of FUS levels on circRNA expression. Nevertheless, there are significant open questions regarding the role of FUS, leaving it unclear whether the protein indeed directly contributes to circRNA biogenesis or whether the apparent phenotypes arise indirectly through disruption of other cellular processes. Considering the increasing number of publications providing unbiased identifications of circRNAs in various physiological models, it is essential that the authors test more directly a role for FUS in circRNAs biogenesis.

Major points

1. Characterization of circRNA

According to the RNAseq data, expression of the circRNAs examined is very low and quite variable between biological replicates. Many circRNAs are detected with only one or two reads. This raises concern regarding their existence and physiological relevance. While the authors performed extensive RT-PCR validations, this approach can often give rise to artifacts, in particular when starting material is low. Thus the authors should (at least for a few circRNA species) validate their existence by a more rigorous approach (northern blot or RNase protection assays).

As also suggested by referee#1, we performed Northern blot assays using probes spanning the back-splicing junction and we tested the expression of the most expressed circRNAs, c-31 and c-78, in ES, GFP⁻-FUS^{+/+}, GFP⁺-FUS^{+/+} and GFP⁺-FUS^{-/-} cells (Fig.2c). Through this approach we excluded the possibility of artifacts and confirmed the down-regulation of these circRNAs in FUS depleted cells as well as their enriched expression in GFP⁺ cells.

2. Role of FUS in circRNA biogenesis

a. Much of the data in the manuscript is correlative, demonstrating alterations in circRNAs when FUS levels are elevated or decreased. Using Clip assays the authors attempt to probe direct association of FUS with segments that undergo circularization. However, these experiments do not strongly support a preferential association of FUS with introns proximal to circularization sites. Only a few candidates were tested and the observed enrichment of FUS binding in these sequences is very modest. The authors should perform a broader analysis by using genome wide analysis of FUS binding sites (some of this data is available) and assess the proportion of FUS-dependent circRNAs whose pre-mature transcripts bind FUS.

We analysed public FUS CLIP data (Lagier-Tourenne et al., 2012 Nat. Neurosci.) and found that intronic regions flanking circRNAs deregulated by FUS KO are enriched in FUS binding sites compared to those flanking unaffected circRNAs. The results of this genome wide analysis are now described in the new Supplementary Fig. 4a and b.

b. Impact of FUS protein level on circRNA expression could be explained by multiple reasons and the causality of altered circRNAs in FUS mutants is unclear. While the data on linear RNAs support that FUS action is independent of transcriptional regulation, other hypotheses cannot be ruled out (impact on circRNA stability, indirect role of FUS through overall modification of molecular repertoire in FUS^{-/-} cells...). To test this more rigorously, the authors could use a minigene reporter approach and assess more directly a role for FUS by mutating its binding site in the reporter.

We raised specific constructs containing the circularizing exon plus ~1500 nucleotides of flanking introns of the c-03 and c-87 host genes and tested their ability to produce circular RNA in the presence or absence of FUS. Fig. 5b shows that the two clones were able to produce the corresponding circRNAs and that this activity was responsive to FUS. In particular, c-03 resulted down-regulated and c-87 up-regulated. These results confirm the involvement of FUS in directing the back-splicing reaction both in a positive and negative way and that ~1500 nucleotides are enough to provide such responsiveness in agreement with CLIP data. Since a clear consensus binding site for FUS is still lacking, the mutational analysis would be quite complex at this stage. We think that the differential behaviour in FUS KD conditions is sufficient at the moment to support our conclusion. Moreover, it cannot be excluded that the effects of FUS could be also mediated by more complex protein-protein-RNA interactions.

3. Figure 2b/c: The expression level of the cognate linear RNAs should be displayed.

As requested, we performed qRT-PCR analysis of the linear forms in GFP⁺-FUS^{+/+} and GFP⁺-FUS^{-/-} cells. The data are shown in the new Supplementary Fig. 2e and f. The results confirm all previous data and show that, despite circRNA modulation, there is no effect on the linear counterpart, therefore indicating that the effects observed are not due to transcriptional control.

Minor points

1. The procedure of RNA sequencing and the data analysis needs to be more clearly described. For example:
- Read length?

A statement indicating read length was added in the Methods section.

- Criteria to consider a read spanning a back-splicing junction (number of nucleotide from each side of the junction)?

We used the "find_circ" pipeline (Memczak et al., Nature 2013) with its default parameters. The pipeline produces two anchors of 20 nucleotides from each end of the analyzed reads and maps them separately, therefore at least 20 nucleotides of each read should be mapped at one side of a head-to-tail splice junction in order to be detected.

- How are the head-to-tail junctions defined (GU/AG dinucleotides)?

After mapping the two anchors, find_circ extends each alignment on the reference genome, until it finds a breakpoint. In order to be annotated as an head-to-tail splice junction, the fragments mapped separately have to reconstitute the entire read and have to contain a GU/AG signal at each breakpoint. We have now included this information in the Materials and Methods section of the paper.

- What statistical analysis was used to define circRNAs differentially regulated between FUS+/- and FUS-/- conditions?

As stated in the Methods section, we used the edgeR software, and in particular glmFIT and glmLRT functions.

2. The meaning of Figure 1b is unclear. The authors attempt to estimate the proportion of circRNAs hosted by coding versus non-coding genes. However, in the text they stated that reads coming from several non-coding genes were discarded in their analysis (line 92). Thus, the numbers in Figure 1b are likely to be misleading.

We only discarded reads mapping linearly to rRNA, tRNA, snRNA, snoRNA and other over-represented non-coding RNAs (this has now been specified as "reads mapping linearly" in the Methods section); since circRNAs were identified based only on the reads mapping head-to-tail, our analysis is theoretically able to find circRNAs hosted by those non-coding RNAs. However, we did not find any circRNA on those genes.

3. Figure 1d: Keeping the same scale for x and y-axis would help the readers to compare the fold-changes of the circRNAs and their cognate linear RNAs.

Figure 1D was changed accordingly.

4. Figure 2: the number of independent biological replicates should be indicated in the figure legends.

We added the numbers of independent biological replicates in each figure legend.

5. The authors state that the nuclear circRNAs c-01, c-087 and c-088 are devoid of intron sequences (line 175). What are the supporting evidences of absence of intronic sequences? It appears that only junctions were analyzed, not the entire sequence of the circRNAs.

The entire sequences of c-01, c-087 circRNAs have been analyzed, confirming the absence of intronic sequences, as shown in Supplementary Figure 2c and d.

REVIEWERS' COMMENTS:

Reviewer #1 (Remarks to the Author):

All my concerns have been adequately addressed.

Reviewer #2 (Remarks to the Author):

In the revised manuscript, the authors performed quantitative RT-PCR analysis and Northern blotting to validate the differential expression of MN enriched circRNA upon FUS deletion. They also re-investigated the effect of ALS linked FUS mutant on circRNA expression in FUS KD N2a cell line to minimize the impact of endogenous FUS. Further examination of selected circRNA in human iPSCs carrying FUS-P525L mutant confirms the results performed in N2a cells and links FUS dependent circRNA production to ALS pathology. Importantly, they analyzed a published FUS CLIP seq data and found that FUS binding sites are enriched in intronic regions flanking circRNAs deregulated by FUS KO. Two artificial reporters containing the exons and flanking intron regions involved in back-splicing event also support the potentially direct involvement of FUS in circRNA biogenesis.

Overall, the authors addressed most of the concerns raised by reviewers and made much improvement of the manuscript. I believe the revised manuscript would be appropriate for publication in Nature Communication if the authors make further revisions as suggested below:

- Fig 2c: why there is no linear form of c-78 detected? Also, to further validate the expressional difference between circRNA/linear RNA, a Northern blots of both linear and circRNAs should be provided.
- Fig S3a shows circRNA expression in differentiated N2a but the main text mentions that these circRNAs are expressed both in proliferation and differentiation condition (line 196). Please make a consistent description
- Fig S3a: there are two major bands of c-78, please indicate which one is c-78.
- Fig S4a and S4b don't match to figure legends.
- Please add a description of N2a tet on system in M&M section.
- Line 174-175: there is a grammar error need to be revised
- Line 208-231 are difficult are comprehend, especially a) the description of FUS mutants' effects due to nuclear level or interaction defect and b) the explanation of why FUS mutants

lose enhancing effects on certain circRNAs while retain inhibitory effects on other circRNAs. Please reorganise and revise the writing to consolidate the finding of FUS involving in circRNA production.

Reviewer #3 (Remarks to the Author):

The authors have addressed the main concerns raised in the review.

Reviewer #2 (Remarks to the Author):

In the revised manuscript, the authors performed quantitative RT-PCR analysis and Northern blotting to validate the differential expression of MN enriched circRNA upon FUS deletion. They also re-investigated the effect of ALS linked FUS mutant on circRNA expression in FUS KD N2a cell line to minimize the impact of endogenous FUS. Further examination of selected circRNA in human iPSCs carrying FUS-P525L mutant confirms the results performed in N2a cells and links FUS dependent circRNA production to ALS pathology. Importantly, they analyzed a published FUS CLIP seq data and found that FUS binding sites are enriched in intronic regions flanking circRNAs deregulated by FUS KO. Two artificial reporters containing the exons and flanking intron regions involved in back-splicing event also support the potentially direct involvement of FUS in circRNA biogenesis.

Overall, the authors addressed most of the concerns raised by reviewers and made much improvement of the manuscript. I believe the revised manuscript would be appropriate for publication in Nature Communication if the authors make further revisions as suggested below:

- Fig 2c: why there is no linear form of c-78 detected? Also, to further validate the expressional difference between circRNA/linear RNA, a Northern blots of both linear and circRNAs should be provided.

The Northern analyses shown in Fig 2C were carried out, as requested in the first round of revision by the same reviewer in order to compare the circular RNA expression in WT versus FUS KO motoneurons (“it would be more convincing to design additional primer sets to validate the change of identified circRNA expression in WT and FUS KO MNs, to verified by Northern Blotting”),

Northern is a suitable procedure to quantify the expression of the same type of molecules (in particular circRNAs) in different conditions. Instead, it is not the proper method to quantify differences between the circular versus linear forms for several reasons: i) the linear forms are in general much longer and might be transferred to the membrane much less efficiently; ii) the probes utilized in the experiments are across the back-splice junction and do not necessarily hybridize efficiently on the linear counterpart (this varies quite a lot from probe to probe depending on their sequence). Indeed, we managed to see the two linear isoforms for c-31 but not for c-78. For these reasons we used, as shown in Fig 2a and Supplementary Fig. 2e, qRT-PCR to discriminate and better quantify the linear versus the circular forms.

- Fig S3a shows circRNA expression in differentiated N2a but the main text mentions that these circRNAs are expressed both in proliferation and differentiation condition (line 196). Please make a consistent description

We corrected the sentence consistently

- Fig S3a: there are two major bands of c-78, please indicate which one is c-78.

We indicated the band corresponding to c-78 by an arrow.

- Fig S4a and S4b don't match to figure legends.

Sorry for the mistake, it has been fixed

- Please add a description of N2a tet on system in M&M section.

We added this in Methods.

- Line 174-175: there is a grammar error need to be revised

We corrected this error.

- Line 208-231 are difficult to comprehend, especially a) the description of FUS mutants' effects due to nuclear level or interaction defect and b) the explanation of why FUS mutants lose enhancing effects on certain circRNAs while retain inhibitory effects on other circRNAs. Please reorganise and revise the writing to consolidate the finding of FUS involving in circRNA production.

We have rephrased the sentence and made it clearer.